# On Adversarial Bias
# and the Robustness of Fair Machine Learning

## Abstract

Optimizing prediction accuracy can come at the expense of fairness. Towards minimizing discrimination against a group, fair machine learning algorithms strive to equalize the error of a model across different groups, through imposing fairness constraints on the learning algorithm. But, are decisions made by fair models trustworthy? How sensitive are fair models to changes in their training data? By giving equal importance to groups of different sizes and distributions in the training set, we show that fair models become more fragile to outliers. We study the trade-off between fairness and robustness, by analyzing the adversarial (worst-case) bias against group fairness in machine learning and by comparing it with the effect of similar adversarial manipulations on regular models. We show that the adversarial bias introduced in training data, via the sampling or labeling processes, can significantly reduce the test accuracy on fair models, compared with regular models. Our results demonstrate that adversarial bias can also worsen a model's fairness gap on test data, even though the model satisfies the fairness constraint on training data. We analyze the robustness of multiple fair machine learning algorithms that satisfy equalized odds (and equal opportunity) notion of fairness.

## 1 Introduction

Trustworthiness is a crucial requirement of machine learning algorithms in critical decision making processes, as highlighted by many AI regulations and policies as well as technical research papers. Algorithmic fairness is at the core of trust requirements for automated decision making in sensitive domains. Group fairness measures, such as equal opportunity and equalized odds (Hardt et al., 2016) which is the focus of this paper, suggest equalizing the model's behavior across groups identified based on a protected attribute (e.g., race) to avoid systemic discrimination against protected groups (Agarwal et al., 2018; Calders et al., 2009; Hardt et al., 2016; Madras et al., 2018).

The main question that we are interested in is whether fair models are trustworthy, and in particular robust with respect to changes to their training data. In this paper, we study if and how **achieving group fairness can increase the susceptibility of machine learning models to a small fraction of adversarially-sampled outliers in the training set**.

A large body of work shows machine learning is vulnerable to noisy and adversarial data (Biggio et al., 2012; Chen et al., 2017; Jagielski et al., 2018; Koh & Liang, 2017; Li et al., 2016; Mei & Zhu, 2015a; Shafahi et al., 2018; Steinhardt et al., 2017; Suciu et al., 2018). Recent work studies the performance of fair machine learning in the presence of noisy training data with under-representation and labeling bias (Blum & Stangl, 2020; Calders & Žliobaitė, 2013; De-Arteaga et al., 2018; Jiang & Nachum, 2020; Kallus & Zhou, 2018; Lamy et al., 2019). Under the assumptions of *uniform* noise and in the theoretical setting of having an unlimited number of training data, these works analyze the effect of noisy/biased training data on fair models. Interestingly, Blum & Stangl (2020) show that ERM with equal opportunity can recover the Bayes-optimal classifier from biased data. In other words, fair algorithms are more robust to certain types of bias in the training dataset than standard learning algorithms (without fairness constraints). However, there has been little quantitative analysis of the interaction between group fairness and robustness of the model under realistic settings (finite training data and non-uniform noise). In this paper, we quantitatively measure the impact of group fairness on the robustness of the model under **worst-case (adversarial) bias**, which exactly aims at minimizing the chance of recovering from biased data. To the best of our knowledge, this paper

provides the first quantitative analysis for the robustness of fair machine learning algorithms in the adversarial setting.

We assume the training data is biased, through adding a small fraction of outliers that are adversarially sampled (and labeled) to degrade the test accuracy of fair models. We exploit the fact that algorithms with group-fairness constraints approximately equalize the influence of different groups, in the training set, on the model. Equality of the *group* influence would consequently change the influence of *individual* data samples across different groups in a disproportionate way due to differences in size and distribution of the groups. Thus, the model's susceptibility to worst-case outliers is largely dependent on how the outliers are distributed across different subgroups.

We extensively evaluate the robustness of fair machine learning on multiple fairness algorithms (Hardt et al., 2016; Agarwal et al., 2018; Rezaei et al., 2020; Cotter et al., 2019; Zhang et al., 2018) and benchmark datasets (Dua & Graff, 2017; Larson et al., 2017; mep; ahr) to investigate how, why, and under what circumstances models with group fairness are more fragile with respect to adversarial bias compared to unconstrained models. We show that group *fairness reduces robustness*. Models trained using various fair machine learning algorithms are all more susceptible to adversarial bias compared with unconstrained models. We can observe this effect even for the case of the most limited scenario of adversarial data *sampling* for a small fraction of the training set, without manipulating the data features and labels. We notice this is because that adversarial bias amplifies the cost of fairness on model accuracy by placing the outliers into the smallest group with the least frequent label. It effectively reduces the best achievable accuracy for the smallest subgroup, limiting the fair models' accuracy on the minority. In this case, the model sacrifices its accuracy over the majority group to satisfy the fairness constraint. It results in a significant accuracy on the overall dataset. Furthermore, we present the potential trade-offs between robustness and fairness. Finally, adversarial manipulation of the training data prevents the model from generalizing its fairness to clean test data even though it *is* guaranteed on training data. This results in models that, according to the fairness measure, are even more discriminatory than unconstrained models, but on a different part of the population.

This work introduces a significant challenge towards designing trustworthy machine learning algorithms. We emphasize that, as shown in our results, the fair models trained under noisy data could be significantly unfair (with respect to the same fairness measures). Thus, sensitivity to changes in data undermines both fairness and accuracy of models. This calls for designing new fairness measures which are not inherently susceptible to noise.

## 2 BACKGROUND AND PROBLEM STATEMENT

**Machine learning.** Consider a classifier $f_\theta : \mathcal{X} \to \mathcal{Y}$, that maps the feature space $\mathcal{X}$ to labels $\mathcal{Y}$. The model is represented by its parameters $\theta$ taken from a parameter space $\Theta$. The model is trained to minimize a loss function $\ell : \Theta \times \mathcal{X} \times \mathcal{Y} \to \mathbb{R}_+$ over its training set $\mathcal{D}$. We let $X$ and $Y$ denote the random variables associated with the features and the labels, and $(X, Y)$ to denote the underlying distribution of the data. We obtain the model parameters by solving $\min_{\theta \in \Theta} \frac{1}{|\mathcal{D}|} \mathcal{L}(\theta; \mathcal{D})$, where $\mathcal{L}(\theta; \mathcal{D}) = \sum_{(x,y) \in \mathcal{D}} \ell(\theta; x, y)$ is the cumulative loss of the model over the training set $\mathcal{D}$.

**Fairness.** We assume all data points are split into groups based on a protected attribute $S \in \mathcal{S}$ (e.g., gender). This attribute could be part of the feature set $\mathcal{X}$. We focus on *equalized odds*, which is a widely-used notion for group fairness (Hardt et al., 2016). [1] Following previous works (Agarwal et al., 2018; Donini et al., 2018), we say a classifier $f_\theta$ is $\delta$-fair under equalized odds if

$$\Delta(\theta, \mathcal{D}) \coloneqq \max_{y \in \mathcal{Y}, a, b \in \mathcal{S}} \left| \Pr_{\mathcal{D}}[f_\theta(X) \neq y | S = a, Y = y] - \Pr_{\mathcal{D}}[f_\theta(X) \neq y | S = b, Y = y] \right| \leq \delta, \quad (1)$$

where the probabilities are computed empirically over the training data set $\mathcal{D}$. We refer to $\Delta$ as the model's empirical **fairness gap**. A model satisfies exact equalized odds fairness when $\delta = 0$. In practice, fairness is usually achieved by ensuring $\delta$-fairness empirically on the model's training set, e.g., through minimizing the model's empirical loss under $\delta$-fairness as a constraint (Agarwal et al., 2018) or post-processing (Hardt et al., 2016). We define the constraint $C(\theta, \mathcal{D}) \coloneqq \Delta(\theta, \mathcal{D}) - \delta \leq 0$ as a **fairness constraint**. We refer to the models learned with the fairness constraint as *fair models*,

---

[1] Extension and analysis for other group fairness metrics (i.e., *equal opportunity* (Hardt et al., 2016)) can be found in Appendix E.7.

to distinguish them from *unconstrained models* that are learned without any fairness constraint. We quantify the performance of a model based on its test accuracy, and its fairness gap on the test dataset.

**Problem statement.** The primary research question we investigate in this paper is whether, how, and why models with (equalized odds) group fairness are less robust to adversarial bias, compared with unconstrained models. We consider a model is robust if changing a small fraction of the training set does not significantly downgrade the predictive power of the model.

Towards quantifying the robustness, we assume the biased training set $\mathcal{D}$ is composed of the clean dataset $\mathcal{D}_c$ of size $n$, and the adversarially chosen dataset $\mathcal{D}_p$ of size $\epsilon n$. The clean training set $\mathcal{D}_c$ and test set $\mathcal{D}_{\text{test}}$ are sampled from the same underlying distribution $(X, Y)$. So, we investigate the effect of the bias in the training set, which is introduced through $\mathcal{D}_p$. We consider two variations of bias: **adversarial sampling**, and **adversarial labeling**. These are the worst-case sampling and labeling bias, where $\mathcal{D}_p$ is chosen to maximize the loss of a model. Let $\mathcal{D}_k$ be a dataset sampled from $(X, Y)$, similar to the clean data. In adversarial sampling, we choose outliers $\mathcal{D}_p \subset \mathcal{D}_k$, and in adversarial labeling, we choose them also with the possibility of crafting their labels $\mathcal{D}_p \subset \{(x, y') : (x, y) \in \mathcal{D}_k, y' \in \mathcal{Y}\}$. Based on this setting, the maximum vulnerability of a fair model in the presence of adversarial bias can be formulated as a bi-level optimization problem subject to the fairness constraint $C(\theta, \mathcal{D}) \le 0$:

$$\max_{\mathcal{D}_p} \mathbb{E}_{(X,Y)}[\ell(\hat{\theta}; X, Y)], \text{ where } \quad \hat{\theta} \coloneqq \operatorname*{argmin}_{\theta \in \Theta} \frac{\mathcal{L}(\theta; \mathcal{D}_c \cup \mathcal{D}_p)}{|\mathcal{D}_c \cup \mathcal{D}_p|}, \text{ st. } C(\theta, \mathcal{D}_c \cup \mathcal{D}_p) \le 0, \quad (2)$$

where the expectation is taken over the underlying (clean) data distribution. The outer maximization searches for the strongest $\mathcal{D}_p$ that maximizes the expected loss of the fair model (given by $\hat{\theta}$). This fair model is obtained by solving the inner constrained minimization on the biased training dataset $\mathcal{D}_c \cup \mathcal{D}_p$. The expected loss can be measured on a test set $\mathcal{D}_{\text{test}}$ sampled from the data distribution $(X, Y)$.

To generate the strongest $\mathcal{D}_p$, we assume the knowledge of the learning algorithm (e.g., logistic regression, SVM) and the clean training dataset $\mathcal{D}_c$ is available when generating $\mathcal{D}_p$, but the exact fair learning algorithm is unknown. It allows us to obtain an upper bound on the performance degradation incurred by the adversarial bias and serves as a starting point towards understanding the maximal vulnerability. Besides, for investigating the effect of fairness constraints on the model robustness, we evaluate the robustness of unconstrained models as a baseline. Adversarial bias against unconstrained models is equivalent to the problem of data poisoning attacks (we discuss more in Section 6).

## 3 ADVERSARIAL BIAS

To find the strongest $\mathcal{D}_p$ for evaluating the robustness of fair models, we need to solve the bi-level optimization problem (2), which is non-convex and intractable (Bard, 1991; Hansen et al., 1992; Deng, 1998). The fairness constraint makes the problem even more difficult.[2] In this section, we explain how to approximate problem (2) to design effective adversarial strategies.

In the problem (2), it is hard to track the influence of $\mathcal{D}_p$ on the test loss as it can only affect the test loss via the model's parameters. To make progress, we first approximate the loss on the test data by the loss on the clean training data, following the approximations used for designing poisoning attacks against unconstrained models (Steinhardt et al., 2017). Specifically, let $\hat{\theta}$ be the solution to the inner optimization problem and we have $\mathcal{L}(\hat{\theta}; \mathcal{D}_{\text{test}})/|\mathcal{D}_{\text{test}}| \approx \mathcal{L}(\hat{\theta}; \mathcal{D}_c)/|\mathcal{D}_c| \le \mathcal{L}(\hat{\theta}; \mathcal{D}_c \cup \mathcal{D}_p)/|\mathcal{D}_c|$. As long as the model has enough capacity to fit but does not overfit the training dataset (which can be achieved by appropriate regularization), the loss on the biased training dataset (i.e., RHS of the inequality) provides a good approximation for the test loss, which allows us to explicitly measure the impact of the poisoning data. Therefore, we replace the objective of the inner minimization in Eq. (2) with $\mathcal{L}(\theta; \mathcal{D}_c \cup \mathcal{D}_p)/n$, where $n = |\mathcal{D}_c|$. The resulting optimization problem is hard to solve due to the bi-level optimization and the constraints in the inner optimization. To resolve this, we then relax the inner constrained optimization by introducing a Lagrange multiplier $\lambda \in \mathbb{R}_+$:

$$\min_{\theta \in \Theta} \left[ \frac{\mathcal{L}(\theta; \mathcal{D})}{n}, \text{ s.t. } C(\theta, \mathcal{D}) \le 0 \right] = \min_{\theta \in \Theta} \max_{\lambda \in \mathbb{R}_+} \left( \frac{\mathcal{L}(\theta; \mathcal{D})}{n} + \lambda C(\theta, \mathcal{D}) \right) \ge \max_{\lambda \in \mathbb{R}_+} \min_{\theta \in \Theta} \left( \frac{\mathcal{L}(\theta; \mathcal{D})}{n} + \lambda C(\theta, \mathcal{D}) \right),$$

---

[2]We would like to point out that, for the unconstrained model, under the convex assumption of the loss function, it is possible to find the approximate solution by replacing the inner optimization with its stationarity (KKT) condition (Biggio et al., 2012; Koh & Liang, 2017).

where $\mathcal{D} = \mathcal{D}_c \cup \mathcal{D}_p$. The inequality follows from the weak duality theorem. We can now find $\mathcal{D}_p$ by maximizing a lower bound provided by the Lagrangian function $\min_{\theta \in \Theta} \left( \mathcal{L}(\theta;\mathcal{D})/n + \lambda C(\theta, \mathcal{D}) \right)$ for a fixed $\lambda \in \mathbb{R}_+$. Indeed, maximizing the lower bound provided by the Lagrangian function would result in a solution with a high loss (which is guaranteed to be at least equal to the loss for the lower bound) for the original problem. In this optimization procedure, we can also replace the fairness constraint $C(\theta, \mathcal{D}) := \Delta(\theta, \mathcal{D}) - \delta$ with the fairness gap $\Delta(\theta, \mathcal{D})$, because the constant value $\delta \geq 0$ does not affect the solution for the Lagrangian. Finally, by considering all the above-mentioned steps, the new optimization problem is:

$$\max_{\mathcal{D}_p} \min_{\theta \in \Theta} \left( \frac{\mathcal{L}(\theta; \mathcal{D}_c \cup \mathcal{D}_p)}{n} + \lambda \Delta(\theta, \mathcal{D}_c \cup \mathcal{D}_p) \right). \tag{3}$$

Thus, to solve the problem (2), the alternative goal is to find $\mathcal{D}_p$ that maximizes a linear combination of the training loss and the model's violation from the fairness constraint, where $\lambda$ controls the penalty for the violation. In the light of those approximations, we design two algorithms to generate $\mathcal{D}_p$.

**An Approximation for the Fairness Gap.** Finding $\mathcal{D}_p$ is an intractable combinatorial optimization problem because the fairness gap could not be split into separate functions of individual data points. It is still hard to track the influence of individual data points. To resolve it, we first find an additive proxy for fairness gap. We substitute the fairness gap $\Delta$ by the average of an approximate contribution of each training data point to the fairness gap. This allows us to design an efficient sequential policy. More specifically, let $\{(x,y)\}^k$ be a multi-set with $k$ repetitions of a data point $(x,y)$. Consequently, $\mathcal{D} \cup \{(x,y)\}^k$ is equivalent to adding $k$ copies of $(x,y)$ to $\mathcal{D}$. In this setting, for any data point $(x,y) \in \mathcal{D}_p$, the fairness gap $\frac{1}{\epsilon n}\Delta\left(\theta, \mathcal{D}_c \cup \{(x,y)\}^{\epsilon n}\right)$ is a proxy for measuring the contribution of that data point to the fairness gap $\Delta(\theta, \mathcal{D}_c \cup \mathcal{D}_p)$. In other words, it measures how the fairness gap changes if $\epsilon n$ copies of $(x,y)$ is added to the clean data. Also, the maximum of $\Delta\left(\theta, \mathcal{D}_c \cup \{(x,y)\}^{\epsilon n}\right)$ over all data points $(x,y) \in \mathcal{D}_p$ provides an upper bound on the fairness gap of the model, when the size of $\mathcal{D}_p$ is $\epsilon n$. Given this proxy for the contribution of each data point to the fairness gap, we obtain the following approximation: $\Delta\left(\theta, \mathcal{D}_c \cup \mathcal{D}_p\right) \approx \sum_{(x,y) \in \mathcal{D}_p} \frac{1}{\epsilon n}\Delta\left(\theta, \mathcal{D}_c \cup \{(x,y)\}^{\epsilon n}\right)$. By substituting the fairness gap with its proxy, now we can solve the following optimization problem:

$$\max_{\mathcal{D}_p} \min_{\theta \in \Theta} \left( \frac{1}{n}\mathcal{L}\left(\theta; \mathcal{D}_c \cup \mathcal{D}_p\right) + \frac{\lambda}{\epsilon n} \sum_{(x,y) \in \mathcal{D}_p} \Delta\left(\theta, \mathcal{D}_c \cup \{(x,y)\}^{\epsilon n}\right) \right) = \max_{\mathcal{D}_p} M(\mathcal{D}_p) := M^* \tag{4}$$

where $M(\mathcal{D}_p)$ is the loss incurred by a poisoning set $\mathcal{D}_p$ on the fair model, and $M^*$ is the maximum loss of the fair model under any choices of $\mathcal{D}_p$. Algorithm 1, a variant of the no-regret online gradient descent methods (Hazan, 2016), presents our solution to problem (4). It initializes a model $\theta^0 \in \Theta$, and identifies $\epsilon n$ points for $\mathcal{D}_p$ iteratively. The feasible set of points $\mathcal{F}(\mathcal{D}_k)$ is determined by the adversarial bias setting. For adversarial sampling, we have $\mathcal{F}(\mathcal{D}_k) = \mathcal{D}_k$, and for adversarial labeling, $\mathcal{F}(\mathcal{D}_k) = \{(x,y') : (x,y) \in \mathcal{D}_k, y' \in \mathcal{Y}\}$. The algorithm iteratively performs the following steps:

- **Data point selection.** (Algorithm 1, line 5): It selects a data point with the highest impact on a weighted sum of the loss function and the fairness gap with respect to the model parameter $\theta^{t-1}$.

- **Parameter update.** (Algorithm 1, line 7): The parameters are updated to minimize the penalized loss function based on the selected data point $(x^t, y^t)$. In this way, the algorithm (through the approximations made by the Lagrange multiplier and the surrogate function) keeps track of the fair model under the set of already selected data points for $\mathcal{D}_p$.

In Theorem 1, by following the approach proposed by (Steinhardt et al., 2017), we relate the performance of Algorithm 1 with the maximum loss in Eq. (4). Moreover, in Appendix C.2, we prove that under some reasonable conditions (e.g., by using similar assumptions made by (Donini et al., 2018) to approximate the fairness gap), our algorithm finds the (nearly) optimal solution for Eq. (4).

**Theorem 1.** *Let $\mathcal{D}_p^*$ be the data set produced by Algorithm 1. Let $\mathrm{Regret}(\epsilon n)$ be the regret of this online learning algorithm after $\epsilon n$ steps. The performance of the algorithm is guaranteed by*

$$M^* - M(\mathcal{D}_p^*) \leq \frac{\mathrm{Regret}(\epsilon n)}{\epsilon n}, \tag{5}$$

*where $M^*$ and $M(\mathcal{D}_p^*)$ are the loss of the fair model under the optimal $\mathcal{D}_p$ and $\mathcal{D}_p^*$, respectively.[3]*

---

[3] The regret of a decision-maker is defined as the difference between the total cost incurred and that of the best-fixed decision in hindsight.

---

**Algorithm 1** Online Gradient Descent Algorithm for Generating $\mathcal{D}_p$ for Fair Models

---

1: **Input:** Clean data $\mathcal{D}_c$, $n = |\mathcal{D}_c|$, feasible set $\mathcal{F}(\mathcal{D}_k)$, $\epsilon n$ (the size of $\mathcal{D}_p$), penalty parameter (Lagrange multiplier) $\lambda$, learning rate $\eta$.
2: **Output:** $\mathcal{D}_p$.
3: Initialize $\theta^0 \in \Theta$
4: **for** $t = 1, \cdots, \epsilon n$ **do**
5:      $(x^t, y^t) \leftarrow \text{argmax}_{(x,y) \in \mathcal{F}(\mathcal{D}_k)} \left[ \epsilon \cdot \ell(\theta^{t-1}; x, y) + \lambda \cdot \Delta \left( \theta^{t-1}, \mathcal{D}_c \cup \{(x,y)\}^{\epsilon n} \right) \right]$
6:      $\mathcal{D}_p \leftarrow \mathcal{D}_p \cup \{(x^t, y^t)\}$
7:      $\theta^t \leftarrow \theta^{t-1} - \eta \left( \frac{\nabla \mathcal{L}(\theta^{t-1}; \mathcal{D}_c)}{n} + \nabla \left[ \epsilon \cdot \ell(\theta^{t-1}; x^t, y^t) + \lambda \cdot \Delta(\theta^{t-1}, \mathcal{D}_c \cup \{(x^t, y^t)\}^{\epsilon n}) \right] \right)$
8: **end for**

---

The proof for the theorem is deferred to Appendix C. From the theorem, it is clear that when the average regret $\text{Regret}(\epsilon n)/\epsilon n$ is small, the Algorithm 1 will result in a nearly optimal $\mathcal{D}_p$.

**A Surrogate Function for the Fair Model.** The parameter update step in Algorithm 1 provides an approximation (through adding the fairness constraint as a penalty and approximating the fairness gap) for the fair model. Differently, our Algorithm 2 approximates the fair model using the unconstrained model. More specifically, Algorithm 2 iteratively adds data points that maximize a combination of the loss and the fairness gap, however, over the unconstrained model. The reason for this approach is that we hope the points with the largest weighted sum of the loss and the fairness gap on the unconstrained model may still have a large weighted sum on the fair models. Algorithm 2 in Appendix C.3 presents the pseudo-code of this algorithm. An advantage of Algorithm 2 is that it reduces the chance of getting stuck in local minima because, in each parameter update, it makes a step towards the negative gradient of the exact unconstrained loss. This is in contrast with Algorithm 1, where due to the difficulty of approximating a constrained max-min problem, it might converge to some parameters not close to the fair model at all. We should point out that the algorithm and objectives are similar to the data poisoning attacks against unconstrained models when $\lambda = 0$ (Steinhardt et al., 2017). In this case, the fairness constraints are not exploited to introduce adversarial bias. In Section 5, we empirically show that $\mathcal{D}_p$ generated by exploiting the fairness gap (i.e., $\lambda > 0$) can incur a higher test loss of fair models compared with the case where $\lambda = 0$.

## 4 EVALUATION SETUP

**Datasets and models.** We conduct experiments on the COMPAS dataset (Larson et al., 2017), Adult dataset (Dua & Graff, 2017), Medical Expenditure Panel Survey (MEPS) dataset (mep; ahr), as well as synthetic data generated with the same setting as in (Zafar et al., 2019). We use the binary protected features in those datasets. We present the results on COMPAS, Adult and synthetic datasets in this section, and the results on all datasets are deferred to Appendix E. On all datasets, we train logistic regression models. The accuracy of classification models on the three real-world datasets is low and close to predicting the most frequent label in the set. This does not help to understand the behavior of models in the presence of adversarial bias. Hence, we perform data pre-processing to separate *hard examples* from the rest of the data. Hard examples are data points with a large loss on a trained model on the entire dataset. We will use hard examples as one of our baselines. We also add hard examples to the attack dataset. We refer the reader to Appendix E.1 for more details.

**Fair machine learning algorithms.** We evaluate the robustness of existing learning algorithms that achieve equalized odds to show that the susceptibility to adversarial bias is a common issue for all of them. Thus, we train logistic regression models with the *equalized odds* fairness constraint, by using the post-processing approach (Hardt et al., 2016), the reductions approach (Agarwal et al., 2018), and fair algorithms proposed in (Zhang et al., 2018; Rezaei et al., 2020; Cotter et al., 2019). Details about these fair algorithms are presented in Appendix D.

**Adversarial bias.** The data points in $\mathcal{D}_p$ are selected from $\mathcal{F}(\mathcal{D}_k)$ using Algorithm 1 (with $\lambda = \epsilon$ on COMPAS and synthetic, and $\lambda = 0.1\epsilon$ on Adult), and Algorithm 2 (with $\lambda = 100\epsilon$). We use Algorithm 2 with $\lambda = 0$ to generate biased data for unconstrained models (which is the same as the algorithm for poisoning attacks against unconstrained models (Steinhardt et al., 2017)). In all the cases, we select points in $\mathcal{D}_p$ without repetition in order to measure the robustness in a realistic setting. See Appendix E.2 for a discussion on choosing $\lambda$ and the implementation details.

Table 1: Test accuracy and fairness gap of unconstrained models and fair models trained on biased training data - COMPAS, Adult and synthetic datasets. When $\epsilon = 0$, the models are trained on $\mathcal{D}_c$. When $\epsilon = 0.1$, we show the accuracy of unconstrained models when the adversarial bias is introduced by Algorithm 2 with $\lambda = 0$ (Steinhardt et al., 2017) and the worst accuracy of fair models when $\mathcal{D}_p$ is generated by either Algorithm 1 or Algorithm 2, along with the corresponding test fairness gap $\Delta$ defined in (1). The relative accuracy drop is shown in parentheses (defined in Section 4). Table 7 in Appendix E.3 shows the all results.

| Dataset | Model | Benign ($\epsilon = 0$) | | Adv. Sampling ($\epsilon = 0.1$) | | Adv. Labeling ($\epsilon = 0.1$) | |
|---|---|---|---|---|---|---|---|
| | | Test Acc | $\Delta_{\text{test}}$ | Test Acc | $\Delta_{\text{test}}$ | Test Acc | $\Delta_{\text{test}}$ |
| Synthetic | Unconstrained | 87.6 | 0.18 | 87.4 (0.2↓) | 0.11 | 85.1 (2.9%↓) | 0.24 |
| | Fair (Hardt et al., 2016) | 80.0 | 0.03 | 77.2 (3.5%↓) | 0.08 | 70.9 (11.%↓) | 0.04 |
| | Fair (Agarwal et al., 2018) | 84.9 | 0.03 | 80.9 (4.7%↓) | 0.07 | 68.0 (20.%↓) | 0.14 |
| | Fair (Rezaei et al., 2020) | 76.1 | 0.02 | 70.2 (7.8%↓) | 0.03 | 58.3 (23.%↓) | 0.02 |
| | Fair (Cotter et al., 2019) | 85.5 | 0.04 | 84.5 (1.2%↓) | 0.09 | 77.8 (9.0%↓) | 0.16 |
| | Fair (Zhang et al., 2018) | 78.1 | 0.11 | 75.0 (4.0%↓) | 0.16 | 63.0 (19.%↓) | 0.27 |
| COMPAS | Unconstrained | 94.3 | 0.21 | 87.6 (7.1%↓) | 0.26 | 84.7 (10.%↓) | 0.28 |
| | Fair (Hardt et al., 2016) | 87.4 | 0.07 | 70.8 (19.%↓) | 0.27 | 68.2 (22.%↓) | 0.25 |
| | Fair (Agarwal et al., 2018) | 93.6 | 0.06 | 73.1 (22.%↓) | 0.37 | 67.7 (28.%↓) | 0.39 |
| | Fair (Rezaei et al., 2020)) | 84.5 | 0.06 | 63.2 (25.%↓) | 0.09 | 60.6 (28.%↓) | 0.09 |
| | Fair (Cotter et al., 2019) | 91.9 | 0.08 | 79.6 (13.%↓) | 0.27 | 76.2 (17.%↓) | 0.31 |
| | Fair (Zhang et al., 2018) | 81.0 | 0.09 | 66.0 (19.%↓) | 0.14 | 64.1 (21.%↓) | 0.12 |
| Adult | Unconstrained | 94.3 | 0.07 | 94.0 (0.3%↓) | 0.06 | 89.3 (5.3%↓) | 0.06 |
| | Fair (Hardt et al., 2016) | 92.7 | 0.03 | 89.6 (3.3%↓) | 0.16 | 81.1 (13.%↓) | 0.09 |
| | Fair (Agarwal et al., 2018) | 93.8 | 0.04 | 91.7 (2.2%↓) | 0.12 | 80.9 (14.%↓) | 0.09 |
| | Fair (Rezaei et al., 2020) | 84.5 | 0.03 | 78.1 (7.6%↓) | 0.03 | 64.9 (23.%↓) | 0.08 |
| | Fair (Cotter et al., 2019) | 92.7 | 0.04 | 90.3 (2.6%↓) | 0.07 | 83.8 (9.6%↓) | 0.44 |
| | Fair (Zhang et al., 2018) | 89.7 | 0.18 | 85.7 (4.4%↓) | 0.28 | 44.9 (50.%↓) | 0.06 |

**Baseline algorithms and robustness evaluation.** Besides comparing with prior data poisoning attacks against unconstrained models (Steinhardt et al., 2017), we consider the following baselines for augmenting the training dataset. *Random sampling*: randomly selecting data points from $\mathcal{D}_k$. *Label flipping*: randomly selecting data points from $\mathcal{D}_k$ and flipping their labels. *Hard examples*: randomly selecting data points from the set of hard examples. To measure the robustness, we compute the relative test accuracy drop of the model trained on biased data versus the benign model ($\epsilon = 0$) trained exclusively on clean data. A smaller drop in relative test accuracy implies stronger robustness.

## 5 EVALUATION RESULTS

In this section, we first quantify and compare the robustness of fair models and unconstrained models, then show how the adversarial bias affects fair models. By examining the impacts of adversarial bias on the majority and minority groups, we identify the causes of the vulnerability of fair models. Finally, we present the trade-off between fairness and robustness and other effects of adversarial bias.

**Fairness deteriorates robustness.** Table 1 presents the test accuracy and fairness gap of unconstrained models and fair models on all the datasets. We observe a **large relative accuracy drop for all fair models on all the datasets** compared with unconstrained models, even in the adversarial sampling setting where data labels are not changed (only clean data are being added but in an adversarially biased manner). On the synthetic dataset, the relative accuracy drop of fair models trained using the reductions approach (Agarwal et al., 2018) is even more than 10 times larger than that of unconstrained models. The results also imply that the robustness issue is more likely caused by fairness notions instead of fair algorithms. Note that our algorithms serve as approximations of the best algorithm for generating $\mathcal{D}_p$. We can expect an even larger accuracy drop for fair models under stronger algorithms for adversarial bias. Besides, Figure 1 compares the test accuracy of unconstrained models and fair models when $\epsilon$ varies. The baselines, e.g., adding randomly selected hard examples, do not have much effect on the test accuracy of fair models. However, when trained

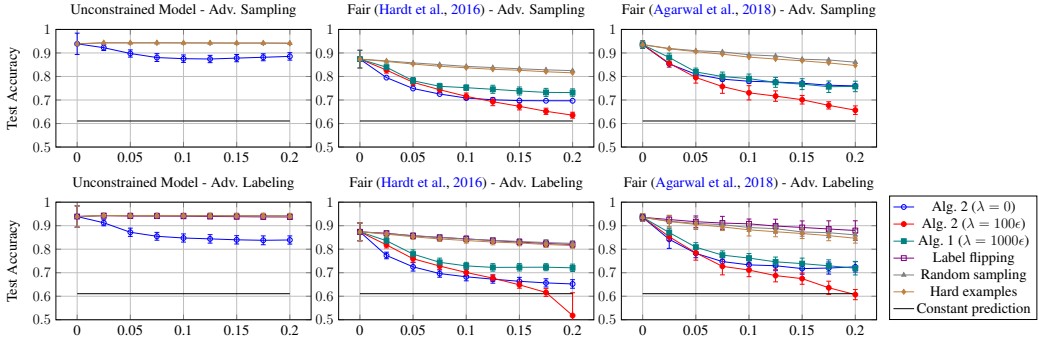

Figure 1: Test accuracy of unconstrained and fair models in the presence of adversarial bias – COMPAS dataset. The x-axis $\epsilon$ is the ratio of the size of $\mathcal{D}_{\mathrm{p}}$ to the size of clean dataset $\mathcal{D}_{\mathrm{c}}$, and reflects the contamination level of the training set. We compare the impact of adversarial bias with baselines and poisoning attacks against unconstrained models, for various $\epsilon$. The difference between test accuracy at $\epsilon = 0$ (benign setting) and a larger $\epsilon$ value reflects the impact of the bias. Constant prediction always outputs the majority label in a clean dataset. The enforced fairness level $\delta$ is 0 and 0.01 for the fair model (Hardt et al., 2016) and fair model (Agarwal et al., 2018), respectively.

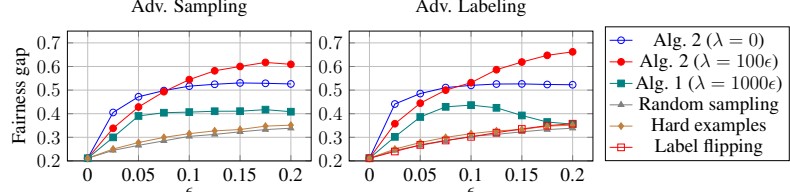

Figure 2: Fairness gap on the unconstrained model with respect to the training data – COMPAS dataset. An unconstrained model is learned on the training data that includes $\mathcal{D}_{\mathrm{p}}$ generated by various algorithms. The fairness gap is defined in (1). The numbers reflect how unfair this unconstrained model is with respect to the protected group on the training data.

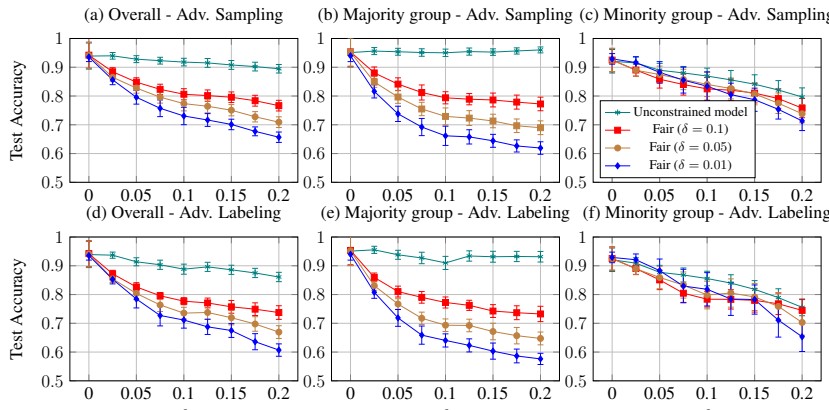

Figure 3: Effect of fairness level $\delta$ on robustness across groups in the presence of adversarial bias – COMPAS dataset. For a given $\epsilon$, training dataset is the same for all the algorithms and $\mathcal{D}_{\mathrm{p}}$ is generated using Alg. 2 with $\lambda = 100\epsilon$. Fair models are trained using reduction approach (Agarwal et al., 2018). The majority group contributes 61% of the test data.

on adversarially biased data, fair models have a much larger relative accuracy drop compared with unconstrained models. At $\epsilon = 0.2$, the test accuracy of fair models approaches what can be achieved even by a constant classifier. These results strongly imply that *fair models are noticeably less robust against adversarial bias than unconstrained models.*

**Adversarial bias amplifies the cost of fairness on model accuracy by increasing the accuracy disparity.** Figures 3(a) & (d) compare the test accuracy of fair models and unconstrained models when trained on the same training dataset. The gap in the accuracy between unconstrained models and fair models reflects the cost of fairness on model accuracy. Notably, as $\epsilon$ increases, this cost increases significantly. For fair model with $\delta = 0.001$, the cost is 0.003 when $\epsilon = 0$ and is increased by more

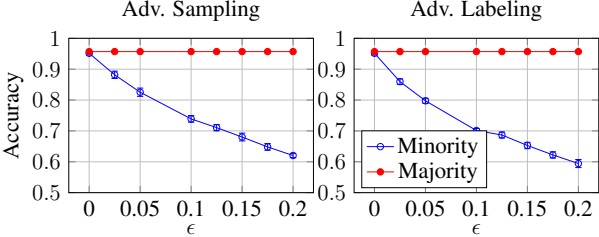

Figure 4: Effect of fairness level $\delta$ on training accuracy across groups in the presence of adversarial bias - COMPAS dataset. For a given $\epsilon$, $\mathcal{D}_p$ is the same for all algorithms (generated using Alg. 2 with $\lambda = 100\epsilon$).

Figure 5: Training accuracy for group-based classifier - COMPAS dataset. We train models only on the majority or minority training data and compare their training accuracy. For a given $\epsilon$, $\mathcal{D}_p$ is generated using Alg. 2 ($\lambda = 100\epsilon$).

than 200 times when $\epsilon = 0.2$ (0.254). It implies that *adversarial bias significantly amplifies the cost of fairness on model accuracy*. We further find that the increase in this cost is positively correlated with the increase in the fairness gap of unconstrained models on training data (which reflects the "unfairness" of unconstrained models on the training data) as shown in Figure 2. It is intuitive that when training accuracy disparity across groups is more significant on unconstrained models, fairness constraints have a stronger impact on model performance, thus incurs a more substantial cost of fairness on model accuracy. In addition, on biased training data generated with Algorithm 2 ($\lambda = 100\epsilon$) at $\epsilon = 0.1$, the fairness gap is 0.54, which is much larger than that of the random sampling baseline (0.3). This indicates that *adversarial bias notably increases the fairness gap, which, in turn, increases the cost of fairness.*

**Implications of adversarial bias for majority vs. minority groups.** Figures 3(b),(c),(e)&(f) show the test accuracy of models on the majority and minority groups. We observe that on fair models, *adversarial bias is significantly more impactful on the majority group*. However, on the unconstrained model, the minority group is the one that incurs a larger loss. We find that the large loss on the minority group for the unconstrained model is mainly caused by $\mathcal{D}_p$. Because the most effective $\mathcal{D}_p$ (generated by Algorithm 2 with $\lambda = 100\epsilon$) mostly belong to the smallest subgroup, i.e., the smallest sensitive group $s$ with the least frequent label $y$ (see the distribution of $\mathcal{D}_p$ across groups in Figure 23 in Appendix E). Consequently, the unconstrained model learns a wrong pattern on the minority group; thus, the accuracy drops significantly for the minority while barely changing on the majority group. On the other hand, the large accuracy drop of fair models on the majority group is primarily due to the fairness constraint. Fair models are enforced to equalize the accuracy across groups on the training dataset. Hence, even though most of the data from the majority group is clean, the training accuracy of fair models on the majority group drops due to the low accuracy on the minority group, as demonstrated in Figure 4. This accuracy drop ultimately translates from train data to test data.

**Adversarial bias increases the inherent "hardness" disparity.** To equalize the accuracy across groups, fair models can increase the accuracy on the minority (the group that has a lower accuracy) or decrease the accuracy on the majority. However, as shown in Figure 4, fair models do not increase the accuracy on the minority but decrease the training accuracy on the majority. We discover that it is related to the best achievable accuracy for two groups (which reflects the inherent "hardness" of classifying positive samples and negative samples). We train group-based (unconstrained) models only using majority or minority training data. The training accuracy approximates the best achievable accuracy on each group for a given training dataset (and for a given type of model). The results are shown in Figure 5. We can see that as $\epsilon$ increases, the best accuracy does not change on the majority group but decreases significantly on the minority (from 0.95 to 0.7 with $\epsilon = 0.2$). Moreover, we find that the training accuracy of fair models on the minority (Figure 4 (b)&(d)) is close to the best

achievable accuracy for the minority (Figure 5). It indicates that *a small fraction of outliers can reduce the best achievable accuracy on the minority significantly*, which limits the increase of the accuracy of fair models on the minority. Consequently, fair models have to sacrifice accuracy on the majority to satisfy the fairness constraint. To conclude, equalizing the accuracy across groups without considering their best achievable accuracy worsens the model's robustness against adversarial bias.

**Trade-off between fairness and robustness.** A fairer model on the training data is less robust against adversarial bias. In Figures 3(a) & (d), fair models with different $\delta$ have similar accuracy when $\epsilon = 0$, (benign setting). As $\epsilon$ increases, the model with a stricter fairness constraint (i.e., smaller $\delta$) has a larger relative accuracy drop. On the other hand, a more robust fair model is less likely to generalize its fairness on the test dataset. In Table 1, the fair algorithm proposed in (Cotter et al., 2019), which is the most robust algorithm against adversarial bias, fails to achieve fairness on the test data. Interestingly, when the unconstrained model is more discriminatory, fair models are less robust. In Table 1, fair models trained using the reduction approach (Agarwal et al., 2018) have a 22% relative accuracy drop on COMPAS dataset, which is 10 times larger than that of fair models trained using the same algorithm on Adult datasets. We observe that this relative accuracy drop is positively correlated with the fairness gap of the unconstrained model in the benign setting ($\epsilon = 0$). It means that when there is more need for a fairness mechanism, fair models are less robust.

**Other effects of adversarial bias.** Fair machine learning aims to achieve fairness for the future data by satisfying fairness constraints on the training data. However, adversarial bias can jeopardize the fairness generalizability of the fair model. In the presence of adversarial bias, the lower the fairness level $\delta$ on training data is, the higher the fairness gap on test data becomes. Moreover, we find that when the adversarial bias is introduced in the training dataset, *the models trained with fairness constraints can become even more discriminatory than unconstrained models.* The detailed results are presented in Appendix E.3. Besides, to approximate the maximal vulnerability, we assume $\mathcal{D}_\mathrm{p}$ is generated with the knowledge about the learning algorithm and clean training dataset. Interestingly, the adversarial bias can still be effective even without this knowledge. See Appendix E.6 for details.

## 6 RELATED WORK AND CONCLUSION

**Related Work** We briefly review the related work here and provide extensive discussion in Appendix B. Adversarial bias against unconstrained models is also known as untargeted data poisoning attacks where the adversary's objective is to degrade the overall test accuracy of the model (Biggio et al., 2012; Mei & Zhu, 2015a; Jagielski et al., 2018; Li et al., 2016; Mei & Zhu, 2015b; Koh et al., 2018; Koh & Liang, 2017; Steinhardt et al., 2017) by adding a small fraction of poisoning data. However, most existing poisoning attacks manipulate the features and labels for the poisoning data. While, in our case, the features of adversarial-chosen data points are clean. Recently, Solans et al. (2021) and Mehrabi et al. (2021b) develop poisoning attacks to increase the "unfairness" of unconstrained models. Differently, our focus is to evaluate the robustness of fair models. Multiple works in the literature study the impact of noisy and biased data on machine learning (Calders & Žliobaitė, 2013; Fogliato et al., 2020; Kallus & Zhou, 2018) but they did not cover adversarial (worst-case) bias. Under varying assumptions, multiple works (Blum & Stangl, 2020; Jiang & Nachum, 2020; De-Arteaga et al., 2018; Lamy et al., 2019; Friedler et al., 2019; Rambachan & Roth, 2019) have proposed strategies to account for under-representation bias and mislabeling while learning models. However, all the above works assume that the noise and bias in the training data is a uniform distribution of under-representation/mislabeling over a subspace of points. Thus, these results may not translate to our case of adversarial (worst-case) bias. Several fair learning algorithms have been proposed to achieve different robustness properties (Wang et al., 2020; Rezaei et al., 2020; Taskesen et al., 2020; Roh et al., 2020; Cotter et al., 2019). However, those algorithms might not achieve robustness in our setting where the adversary controls the labels of a small fraction of samples, and the learner has no additional knowledge about the clean dataset.

**Conclusion** We have introduced adversarial bias against fair machine learning for quantifying the robustness of models with group fairness. Our algorithms exploit the tension between fairness and accuracy and the fact that fair models try to equalize the accuracy on groups with different sensitive attributes. Our experiments show that adding a small percentage of adversarially sampled/labeled data points to the training set can significantly reduce the best accuracy fair models can get on the minority. As the results, a small fraction of the adversarially sampled/labeled data points reduces the model accuracy beyond what they can impose on unconstrained models.

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

## LIST OF TABLES

## LIST OF FIGURES

# A  TABLE OF NOTATIONS

Table 2: List of Notations

| Symbol | Description | Where it is defined |
|---|---|---|
| $\mathcal{X}$ | Features space | Section 2 |
| $\mathcal{Y}$ | Label space | Section 2 |
| $X$ | Random variable associated with features | Section 2 |
| $Y$ | Random variable associated with lables | Section 2 |
| $(X, Y)$ | Underlying distribution of the data | Section 2 |
| $\mathcal{D}$ | Training dataset | Section 2 |
| $N$ | Size of the training dataset $\mathcal{D}$ | Section 2 |
| $\mathcal{D}_c$ | Clean training dataset | Section 2 |
| $n$ | Size of the clean training dataset | Section 2 |
| $\mathcal{D}_p$ | The adversarially-chosen poisoning dataset | Section 2 |
| $\epsilon$ | The ratio of the size of $\mathcal{D}_p$ over the size of clean data in the training set | Section 2 |
| $\Pr_{\mathcal{D}}$ | Computing a probability empirically over a dataset $\mathcal{D}$ | Section 2 |
| $\mathcal{D}_k$ | A set of data points where the feature vectors of $\mathcal{D}_p$ are sampled from | Section 2 |
| $\mathcal{D}_{test}$ | Test dataset | Section 2 |
| $S$ | Sensitive/protected attribute | Section 2 |
| $\mathcal{S}$ | Sensitive/protected attribute Space | Section 2 |
| $\delta$ | Guaranteed fairness level on training data | Section 2 |
| $\Delta$ | Fairness gap | Section 2 |
| $\Delta_{EO}$ | Fairness gap under equalized odds | Section 2 |
| $C(\theta, \mathcal{D})$ | Fairness constraint of $f_\theta$ on dataset $\mathcal{D}$ | Section 2 |
| $\theta$ | Model parameters | Section 2 |
| $\Theta$ | Parameter space | Section 2 |
| $f_\theta$ | Classification model parameterized by $\theta$ | Section 2 |
| $\ell(\theta; x, y)$ | Loss of $f_\theta$ on data $(x, y)$ | Section 2 |
| $\mathcal{L}(\theta; \mathcal{D})$ | Cumulative loss of $f_\theta$ on dataset $\mathcal{D}$ | Section 2 |
| $\hat{\theta}$ | Optimal model parameters trained on $\mathcal{D}$ with fairness constrained | Eq. (2) |
| $\lambda$ | Lagrange multiplier (penalty parameter) | Section 3 |
| $\mathcal{D} \cup \{(x, y)\}^k$ | Adding $k$ copies of the data point $(x, y)$ to set $\mathcal{D}$ | Section 3 |
| $M^*$ | Maximum loss under the optimal attack | Eq. (4) |
| $M(\mathcal{D}_p)$ | The imposed loss by poisoning dataset $\mathcal{D}_p$ on the fair model | Eq. (4) |
| $(x^t, y^t)$ | The data point selected by Algorithm 1 at step $t$ | Section 3 |
| $\theta^t$ | The model parameter chosen by Algorithm 1 at step $t$ | Section 3 |
| $\mathcal{F}(\mathcal{D}_k)$ | Feasible set of $\mathcal{D}_p$ | Section 3 |
| $\eta$ | Learning rate | Algorithm 1 |
| $\mathcal{D}_p^*$ | The dataset produced by Algorithm 1 for introducing adversarial bias into the training dataset | Section 3 |
| Regret(T) | Regret of Algorithm 1 after $T$ steps | Section 3 |
| $U(\theta)$ | Loss of model $f_\theta$ under the optimal attack | Appendix C.1 |
| $U^*$ | Minimum loss under the optimal adversarial bias $\min_{t=1}^T U(\theta^t)$ | Appendix C.1 |
| $g_t(\theta)$ | The loss function for online learning algorithm at step $t$ | Appendix C.1 |
| $\tilde{\theta}$ | Optimal model parameters for minimizing the cumulative loss $\sum_{t=1}^T g_t(\theta)$ | Appendix C.1 |
| $\eta_t$ | Learning rate at time $t$ in Algorithm 1 used in the proof of Corollary 1 | Appendix C.1 |
| $d$ | Upper bound on the diameter of $\Theta$ | Appendix C.1 |
| $G$ | Upper bound on the norm of the subgradients of $g_t$ over $\Theta$ | Appendix C.1 |
| $\tilde{\Delta}(\theta, \mathcal{D})$ | The convex relaxation for the fairness gap of equalized odds | Appendix C.2 |
| $\ell_l(\theta; x, y)$ | Linear loss of model $f_\theta$ on data point $(x, y)$ | Appendix C.2 |
| $\mathcal{D}^{y,s}$ | A set of data points from group $s$ with label $y$ in $\mathcal{D}$ | Appendix C.2 |
| $n^{y,s}$ | The number of data points in $\mathcal{D}^{y,s}$ | Appendix C.2 |
| $R^{y,s}(\theta, \mathcal{D})$ | Average linear loss of $f_\theta$ for data points in $\mathcal{D}^{y,s}$ | Appendix C.2 |
| $\Delta_{EOpp}$ | Fairness gap under equal opportunity | Appendix E.7 |

# B RELATED WORK

In this paper, we aim to analyze the effect of achieving group fairness on the robustness of the model against the adversarially chosen outliers. Thus, in the following, we review the fair machine learning, the robustness of the unconstrained model against outliers, and the recent study about learning fair models under the noisy training data, along with the studies about the trade-off between accuracy and fairness.

**Fairness in Machine Learning**   A classifier that is learned by minimizing the overall cumulative loss might not perform well on one sensitive group (usually the minority group) when the distribution of features per each class is different across groups. In order to address this problem, multiple definitions of fairness are proposed in the literature. Examples include metric equality across sensitive groups (Hardt et al., 2016; Calders et al., 2009), individual fairness (Dwork et al., 2012), causality (Kusner et al., 2017), and many techniques to satisfy group-based fairness (which is the focus of this paper) such as pre-processing methods (Zemel et al., 2013; Madras et al., 2018), in-processing methods (Kamishima et al., 2011; Zafar et al., 2017a;b; Agarwal et al., 2018), and post-processing methods (Hardt et al., 2016). Pre-processing methods aim at finding a new representation of data such that it retains as much information of input features as possible, except those which can lead to bias. In-processing methods enforce fairness during the training process, for example, by incorporating the fairness constraints into the objective function as a regularization term. Post-processing methods correct the predictions of a given trained model, without modifying the training data or the training process. A slightly different approach from these three to achieve fairness is adversarial training (Zhang et al., 2018), where the learning objective is formulated as a game between two players with one player trying to optimize for accuracy and the other strives to achieve fairness. Please refer to (Mehrabi et al., 2021a) for a recent survey on methods to achieve fairness. In this work, we focus on the notion of Equalized odds (Hardt et al., 2016) and train fair models using in-processing approaches (Agarwal et al., 2018; Cotter et al., 2019; Rezaei et al., 2020; Zhang et al., 2018), post-processing (Hardt et al., 2016).

**Data poisoning attack**   Adversarial bias against unconstrained models is also known in the literature as data poisoning attacks when the appearance of an adversary in the system is emphasized. Machine learning systems are susceptible to data poisoning attacks. The adversary's objective is to degrade the overall test accuracy of the model (Biggio et al., 2012; Mei & Zhu, 2015a; Jagielski et al., 2018; Li et al., 2016; Mei & Zhu, 2015b; Koh et al., 2018; Koh & Liang, 2017) or to increase the loss on specific test data points or small sub-populations (Gu et al., 2017; Chen et al., 2017; Burkard & Lagesse, 2017; Shafahi et al., 2018; Koh & Liang, 2017; Suciu et al., 2018). In our paper, we study the impact of adversarial bias on the overall accuracy of fair models without considering the role of an adversary (capacity, knowledge, etc). Steinhardt et al. (2017) propose an optimal algorithm for poisoning attacks on (unconstrained) convex models, given a set of feasible poisoning data points. The algorithm relies on the assumption that test loss of the target model can be approximated as training loss of the model on clean data (assuming $\mathcal{D}_{\text{test}}$ is drawn from the same distribution as the clean training data $\mathcal{D}_{\text{c}}$). Our algorithm is inspired by this work and uses the same online learning framework. Note that, when $\lambda = 0$ in Algorithm 2, it is equivalent to the algorithm in (Steinhardt et al., 2017).

In our setting of adversarial sampling bias, both the feature vectors and the labels are not modified. In *clean-label* data poisoning attacks (Shafahi et al., 2018), the attacker manages to reduce the accuracy of target examples via injecting the correctly labeled data with *modified features*. Another typical data poisoning attack allows the attacker to change both features and labels of the poisoning data. This is first studied in the context of SVMs by Biggio et al. (2012), and has subsequently been extended to linear and logistic regression (Mei & Zhu, 2015b), topic modeling (Mei & Zhu, 2015a), collaborative filtering (Li et al., 2016), and neural networks (Koh & Liang, 2017). In our setting, we assume the features cannot be modified, as we focus on the most practical scenario in decision-making processes that move toward automation. An interesting future direction would be to allow changes of features and introduce adversarial bias using the gradient-based algorithm. Given a larger space for finding the adversarial bias, it is likely that we would observe a larger accuracy drops for fair models.

Instead of reducing the accuracy, Solans et al. (2021) and Mehrabi et al. (2021b) develop poisoning attacks where the attacker's objective is to increase the disparity in the performance of the model

across different groups i.e., increase the unfairness. The key difference in our work is that adversarial bias is introduced to degrade test accuracy of fair models, whereas the algorithms in (Solans et al., 2021; Mehrabi et al., 2021b) degrade the fairness of models trained without any fairness constraints.

**Learning Fair Models from Noisy Training Data**    In most practical scenarios, the training data used for learning models might be biased (under-representation bias) and/or noisy (with mis-labeling). The mis-labeling phenomenon can be random or adversarial. Mislabeling can be seen as a specific case of adversarial labeling bias, where labels of data points from only a certain part of the population are flipped with some probability. Similarly, under-representation bias can be considered as a specific case of adversarial sampling bias. Multiple works in the literature study the impact of noisy and biased data on machine learning.

Calders & Žliobaitė (2013) show that learning an unconstrained model from training data with under-representation and mislabeling bias results in biased predictions on test data. Fogliato et al. (2020) show the effect of noise in labels on predictive bias and the evaluation of fairness metrics of the unconstrained model without any assumption on the structure of noise. Kallus & Zhou (2018) consider the case of systematic censoring in training data (a form of sampling bias) due to which, the classifier that seeks to achieve fairness by equalizing fairness metrics across sensitive groups can still be unfair on the population. We also show a similar result in Table 1 that learning models on training data with adversarial bias increases their fairness gap on the test data.

Under varying assumptions, multiple works (Blum & Stangl, 2020; Jiang & Nachum, 2020; De-Arteaga et al., 2018; Lamy et al., 2019; Friedler et al., 2019; Rambachan & Roth, 2019) have proposed strategies to account for under-representation bias and mislabeling while learning models. De-Arteaga et al. (2018) study selective label bias, where true outcomes corresponding to a certain label cannot be learned, as such examples cannot be added to the training data. The authors propose a method for augmenting the dataset with human expert predictions to mitigate this kind of bias. Jiang & Nachum (2020) propose a re-weighting strategy for recovering the optimal classifier on unbiased data from training data with labeling bias. When uniform random noise is present in the sensitive attribute, it is shown that demographic parity (DP) gap of a fair classifier on test data increases (Lamy et al., 2019). The authors propose a method to compute the exact level of DP gap that needs to be imposed on training data, for achieving a target DP gap on the test data. Blum & Stangl (2020) consider a training set corrupted by under-representation and/or labeling bias. Assuming access to an infinite number of samples and learning different classifiers for different sensitive groups, this work shows that ERM with Equal Opportunity constraint on the biased data can recover the Bayes-optimal classifier for the true data distribution. Friedler et al. (2019) show a slight variation in the fraction of examples from each group can affect the performance of a fair classifier. Rambachan & Roth (2019) consider a specific sampling bias against a group which results in the selection of more samples from that group in the training data. In this case, the more biased the decision-maker is against a group, the more the algorithmic decision rule favors that group.

All the above works (Blum & Stangl, 2020; Jiang & Nachum, 2020; De-Arteaga et al., 2018; Lamy et al., 2019; Friedler et al., 2019; Rambachan & Roth, 2019) assume that the noise and bias in the training data is a uniform distribution of under-representation/mislabeling over a subspace of points and study the consequences of learning from training data with such bias. These results cannot translate to our case of adversarial bias as we consider non-uniform bias over the input space, and bias is introduced with the specific intention of reducing test accuracy.

Several fair learning algorithms have been proposed to achieve different robustness properties. Wang et al. (2020) and Celis et al. (2021) propose algorithms robust to noise in the sensitive attribute, which violates our assumption on unchanged feature vectors. Distributionally robust algorithms studied in (Rezaei et al., 2020; Taskesen et al., 2020) output classifiers that are fair when the true data distribution is slightly different from the estimated one. These approaches might fail to defend in the worst-case adversarial bias as the assumptions on data distributions are not satisfied; we show this observation for the algorithm proposed in (Rezaei et al., 2020) in the evaluation. Similarly, Mandal et al. (2020) study fair and robust algorithms that minimize a distributionally robust loss and are fair with respect to a class of distributions. This class of distributions are weighted perturbations of the training samples. Roh et al. (2020) propose a GAN-based robust and fair learning algorithm assuming an additional clean dataset is available. It aims to achieve robustness and fairness at the same time but lacks theoretical guarantees for both robustness and fairness. For a better generalization of fair

algorithms, Cotter et al. (2019) propose to use a separate validation set during the training process to measure and account for fairness. From Table 1, we observe that this algorithm has a smaller accuracy drop in the presence of adversarial bias yet fails to be fair on test data. Nanda et al. (2021) study robust and fair machine learning algorithm with respect to the adversarial perturbation at the test time which differs from our setting. The concurrent work by Roh et al. (2021) proposes a robust and fair machine learning algorithm that might be used to solve the issues raised by our paper.

**Fairness-Accuracy Trade-offs**   Imposing fairness constraints might come at a cost of the model's performance. The effect of fair classification on accuracy and the compatibility of various definitions with each other have been studied in some related works (Corbett-Davies et al., 2017; Kleinberg et al., 2017). Kleinberg et al. (2017) show that it is *impossible* to achieve equal calibration, false positive rate, and false negative rate if the fraction of positive labeled examples is different across sensitive groups. Corbett-Davies et al. (2017) show that the optimal decision rule is different from the fair decision rules that satisfy fairness definitions (statistical parity, conditional statistical parity, predictive equality). Thus, imposing fairness constraints has a cost on model accuracy. In Section 5, we show that adversarial bias amplifies the cost of fairness. We believe that this observation would motivate further study about the trade-off between fairness and accuracy in the worst-case bias setting.

Under certain assumptions on bias in data and the underlying data distribution, fairness and accuracy are not always in tension as discussed in (Wick et al., 2019; Dutta et al., 2020). These works (Wick et al., 2019; Dutta et al., 2020) argue that the reason for the seeming existence of a trade-off between fairness and accuracy is due to measuring the wrong or biased datasets (which is the only information that is available). Had there been an unbiased dataset available, classifiers could achieve both fairness and accuracy simultaneously without any trade-off. The sampling bias and labeling bias are particularly highlighted in (Wick et al., 2019) and it is mentioned that measuring the fairness and accuracy on datasets with such bias will reflect a trade-off. It was shown in (Dutta et al., 2020) that there exists a distribution for which classifier learned on (biased) observed data can be both perfectly fair and accurate. In our paper, we do not have any assumptions about bias already existing in the data set. We introduce adversarial bias on the given training data set (potentially already biased) and study the robustness of the fair models in the presence of the adversarial bias setting by measuring the relative accuracy drop on the test data set (without adversarial bias).

## C SUPPLEMENTARY THEORETICAL RESULTS

### C.1 PROOF FOR THEOREM 1

*Proof.* We should point out that in this proof we follow the approach of (Steinhardt et al., 2017). Assume $T = \epsilon n$ is the time horizon. We have $\mathcal{D}_{\mathrm{p}}^* = \{(x^1, y^1), \cdots (x^T, y^T)\}$ as the data set produced by Algorithm 1. Also, $\theta^t$ is the parameter chosen by the algorithm at the $t$-th step. First, from max–min inequality we have:

$$M^* \overset{\text{def}}{=} \max_{\mathcal{D}_{\mathrm{p}}} \min_{\theta} \left[ \frac{1}{n}\mathcal{L}(\theta; \mathcal{D}_{\mathrm{c}} \cup \mathcal{D}_{\mathrm{p}}) + \frac{\lambda}{\epsilon n} \cdot \sum_{(x,y)\in\mathcal{D}_{\mathrm{p}}} \Delta\left(\theta, \mathcal{D}_{\mathrm{c}} \cup \{(x,y)\}^{\epsilon n}\right) \right]$$

$$\leq \min_{\theta} \max_{\mathcal{D}_{\mathrm{p}}} \left[ \frac{1}{n}\mathcal{L}(\theta; \mathcal{D}_{\mathrm{c}} \cup \mathcal{D}_{\mathrm{p}}) + \frac{\lambda}{\epsilon n} \cdot \sum_{(x,y)\in\mathcal{D}_{\mathrm{p}}} \Delta\left(\theta, \mathcal{D}_{\mathrm{c}} \cup \{(x,y)\}^{\epsilon n}\right) \right] .$$

Furthermore, for a given $\theta$ we define:

$$U(\theta) \overset{\text{def}}{=} \max_{\mathcal{D}_{\mathrm{p}}} \left[ \frac{1}{n}\mathcal{L}(\theta; \mathcal{D}_{\mathrm{c}} \cup \mathcal{D}_{\mathrm{p}}) + \frac{\lambda}{\epsilon n} \cdot \sum_{(x,y)\in\mathcal{D}_{\mathrm{p}}} \Delta\left(\theta, \mathcal{D}_{\mathrm{c}} \cup \{(x,y)\}^{\epsilon n}\right) \right]$$

$$= \frac{1}{n}\mathcal{L}(\theta; \mathcal{D}_{\mathrm{c}}) + \max_{(x,y)\in\mathcal{F}(\mathcal{D}_{\mathrm{k}})} \left[\epsilon \cdot \ell(\theta; x, y) + \lambda \cdot \Delta\left(\theta, \mathcal{D}_{\mathrm{c}} \cup \{(x,y)\}^{\epsilon n}\right)\right] .$$

We define $U^* = \min_{t=1}^T U(\theta^t)$. Note that for any given $\theta$, we have $M^* \leq U(\theta)$. More specifically, we have $M^* \leq U^*$.

From the definition of $M^*$, for any set, including $\mathcal{D}_{\mathrm{p}}^*$, we have

$$\min_{\theta} \left[ \frac{1}{n}\mathcal{L}(\theta; \mathcal{D}_{\mathrm{c}} \cup \mathcal{D}_{\mathrm{p}}^*) + \frac{\lambda}{\epsilon n} \cdot \sum_{(x,y)\in\mathcal{D}_{\mathrm{p}}^*} \Delta\left(\theta, \mathcal{D}_{\mathrm{c}} \cup \{(x,y)\}^{\epsilon n}\right) \right] = M(\mathcal{D}_{\mathrm{p}}^*) \leq M^*$$

Let us define $T$ different functions

$$g_t(\theta) = \frac{1}{n}\mathcal{L}(\theta; \mathcal{D}_{\mathrm{c}}) + \epsilon \cdot \ell(\theta; x^{t+1}, y^{t+1}) + \lambda \cdot \Delta(\theta, \mathcal{D}_{\mathrm{c}} \cup \{(x^{t+1}, y^{t+1})\}^{\epsilon n}) , \qquad (6)$$

for $0 \leq t \leq T$. Let us define

$$\tilde{\theta} = \operatorname*{argmin}_{\theta\in\Theta} \sum_{t=1}^{T} g_t(\theta) .$$

Note that we have

$$\frac{\sum_{t=1}^{T} g_t(\tilde{\theta})}{T} = M(\mathcal{D}_{\mathrm{p}}^*) \leq M^* \leq U^* \leq \frac{\sum_{t=1}^{T} g_t(\theta^t)}{T}$$

Finally, from the definition of regret we have:

$$\frac{\sum_{t=1}^{T} g_t(\theta^t)}{T} - \frac{\sum_{t=1}^{T} g_t(\tilde{\theta})}{T} = \frac{\mathrm{Regret}(T)}{T} ,$$

which consequently completes the proof of the theorem. □

### C.2 OPTIMALITY CONDITIONS FOR THE SOLUTION OF PROBLEM (4)

In this section, we show under which conditions, our algorithm finds the (nearly) optimal solution for problem (4). We first state a direct consequence of Theorem 1 for a no-regret algorithm which results from a convexity assumption for functions $g_t(\theta)$. We then explain under what conditions this convexity assumption is valid.

**Corollary 1.** *Under the assumption that (i) loss function $\ell$ is convex in $\theta$, (ii) $\Delta(\theta, \mathcal{D})$ is convex in $\theta$ and (iii) $\eta_t = \frac{d}{G\sqrt{t}}$ for $1 \leq t \leq T$, Algorithm 1 produces the near optimal dataset $D_{\mathrm{p}}^*$, such that*

$$M^* - M(D_{\mathrm{p}}^*) \leq \frac{3Gd}{\sqrt{\epsilon n}} \tag{7}$$

*where $\eta_t$ is step size at time $t$, $d$ is an upper bond on the diameter of $\Theta$, and $G$ is an upper bound on the norm of the subgradients of $g_t$ over $\Theta$, i.e., $\|\nabla g_t(\theta)\| \leq G$.*

*Proof.* First note that Algorithm 1 exactly runs as online gradient descent algorithm for $g_t(\theta)$ functions. From the assumptions (i) and (ii), we conclude that functions $g_t(\theta)$ are convex. The theoretical guarantee of the online gradient descent algorithm for convex functions (Hazan, 2016) allows us to bound the average regret

$$\frac{\mathrm{Regret}(T)}{T} \leq \frac{3Gd}{\sqrt{T}} \quad ,$$

where $d$ is an upper bond on the diameter of $\Theta$, and $G$ is an upper bound on the norm of the subgradients of $g_t$ over $\Theta$, i.e., $\|\nabla g_t(\theta)\| \leq G$. Finally, the proof is concluded from this bound for the regret and the result of Theorem 1. $\qquad\square$

Next, we discuss the optimality conditions for linear classifiers with a convex loss, e.g., $\ell(\theta; x, y) = \max(0, 1 - y\langle \theta, x \rangle)$ for SVM. In our paper, we focus on *equalized odds* which is non-convex. We adopt simplification proposed by Donini et al. (2018) to reach convex relaxations of loss and fairness constraint. Instead of balancing prediction error, Donini et al. (2018) propose a fairness definition as balancing the risk among two sensitive groups. Following the same idea, we define the linear loss as $\ell_l$ (e.g., $\ell_l = (1 - f_\theta(x))/2$ for SVM). Based on the linear loss, the convex relaxation for the fairness gap of equalized odds is defined as follows:

$$\tilde{\Delta}(\theta, \mathcal{D}) := \frac{|R^{+,a}(\theta, \mathcal{D}) - R^{+,b}(\theta, \mathcal{D})| + |R^{-,a}(\theta, \mathcal{D}) - R^{-,b}(\theta, \mathcal{D})|}{2} \quad , \tag{8}$$

for $R^{y,s}(\theta, \mathcal{D}) = \frac{1}{n^{y,s}} \sum_{(x,y) \in \mathcal{D}^{y,s}} \ell_l(\theta; x, y)$ where $\mathcal{D}^{y,s}$ is the set of data points from group $s$ with label $y$ in $\mathcal{D}$ and $n_{y,s} = |\mathcal{D}^{y,s}|$. To find the optimal $\mathcal{D}_{\mathrm{p}}$ for the EO fair model, in Eq. (4), we replace loss $\ell$ with a convex loss (e.g. Hinge loss) $\ell_c$ and replace $\Delta(f_\theta; \mathcal{D})$ with the convex relaxation $\tilde{\Delta}(\theta; \mathcal{D})$. Hence, Algorithm 1 produces the nearly optimal data set $\mathcal{D}_{\mathrm{p}}^*$ such that it has the maximal damage on the fair model under our approximations.

As a future research direction, one could try to design new online algorithms that achieve small regrets in the non-convex setting or under better approximations of the fairness constraint. Our framework can then utilize such online algorithms to further investigate the effect of adversarial bias on the robustness of models with fairness constraints.

### C.3 PSEUDOCODE FOR THE ALGORITHM FROM SECTION 3

For the sake of completeness, we present the full pseudo-code for our algorithm proposed in Section 3.

---

**Algorithm 2**

---

1: **Input:** Clean data $\mathcal{D}_{\mathrm{c}}$, $n = |\mathcal{D}_{\mathrm{c}}|$, feasible set $\mathcal{F}(\mathcal{D}_{\mathrm{k}})$, $\epsilon n$ (the size of $\mathcal{D}_{\mathrm{p}}$), penalty parameter (Lagrange multiplier) $\lambda$, learning rate $\eta$.
2: **Output:** $\mathcal{D}_{\mathrm{p}}$.
3: Initialize $\theta^0$
4: **for** $t = 1, \cdots, \epsilon n$ **do**
5: $\quad (x^t, y^t) \leftarrow \mathrm{argmax}_{(x,y) \in \mathcal{F}(\mathcal{D}_{\mathrm{k}})} \left[ \epsilon \cdot \ell(\theta^{t-1}; x, y) + \lambda \cdot \Delta\left(\theta^{t-1}, \mathcal{D}_{\mathrm{c}} \cup \{(x,y)\}^{\epsilon n}\right) \right]$
6: $\quad \mathcal{D}_{\mathrm{p}} \leftarrow \mathcal{D}_{\mathrm{p}} \cup \{(x^t, y^t)\}$
7: $\quad \theta^t \leftarrow \theta^{t-1} - \eta \left( \frac{\nabla \mathcal{L}(\theta^{t-1}; \mathcal{D}_{\mathrm{c}})}{n} + \epsilon \cdot \nabla \ell(\theta^{t-1}; x^t, y^t) \right)$
8: **end for**

---

Table 3: Distribution of data points in clean training dataset and $\mathcal{D}_k$ – COMPAS dataset.

| | Clean | | | $\mathcal{D}_k$ | |
|---|---|---|---|---|---|
| | $y = -$ | $y = +$ | | $y = -$ | $y = +$ |
| $s = 0$ | 28.5% | 31.8% | $s = 0$ | 29.0% | 31.1% |
| $s = 1$ | 32.5% | 7.2% | $s = 1$ | 16.0% | 23.9% |

Table 4: Distribution of data points in clean training dataset and $\mathcal{D}_k$ – Adult dataset.

| | Clean | | | $\mathcal{D}_k$ | |
|---|---|---|---|---|---|
| | $y = -$ | $y = +$ | | $y = -$ | $y = +$ |
| $s = 0$ | 48.5% | 16.5% | $s = 0$ | 45.0% | 23.4% |
| $s = 1$ | 32.3% | 2.6% | $s = 1$ | 27.2% | 4.4% |

## D  FAIR MACHINE LEARNING ALGORITHMS

The **post-processing approach** is the first proposed algorithm to achieve equalized odds (Hardt et al., 2016). The fair model is obtained by adjusting a trained unconstrained model so as to remove the discrimination according to equalized odds. The outcome of this approach is a randomized classifier that assigns to each data point a probability of changing the prediction output by the unconstrained model, conditional on its protected attribute, and predicted label. These probabilities are computed by a linear program that optimizes the expected loss.

Many methods have been proposed to achieve fairness in machine learning (see (Mehrabi et al., 2021a) for a recent survey). The **reductions approach** proposed by (Agarwal et al., 2018) trains a fair randomized classifier over a hypothesis class by reducing the constrained optimization problem to learning a sequence of cost-sensitive classification models. Cost-sensitive classification is used in this as an oracle to solve classification problems resulted from a two-player game: one player (primal variables) minimizes the loss function; the other player (dual variables) maximizes the fairness violation (constraints). Similarly, the fair learning algorithm proposed in (Cotter et al., 2019) is also based on a two-player game. One player optimizes the model parameters on a training dataset to minimize the loss, and the other player enforces the constraints on an independent validation dataset instead of training data. Different from the fair algorithms based on the two-play game, adversarial debiasing (Zhang et al., 2018) is proposed to learn a classifier to maximize prediction accuracy and simultaneously reduce an adversary's ability to determine the protected attribute from the predictions. This approach leads to a fair classifier as the predictions cannot carry any group discrimination information that the adversary can exploit. (Rezaei et al., 2020) trains a fair classifier that is (distributionally) robust when the true data distribution is different from the distribution estimated on training data but their statistics match. The method is applied particularly for classifier using logarithmic loss, where due to the strong duality of the optimization problem, a closed-form solution can be derived.

## E  SUPPLEMENTARY EXPERIMENTAL RESULTS

For the following section, we present the detailed experimental results on COMPAS, Adult, Medical Expenditure Panel Survey (MEPS) and synthetic datasets. All the results on COMPAS dataset are averaged over 100 runs with different random seeds. On Synthetic, MEPS and Adult datasets, all the results are averaged over 50 runs with different random seeds.

### E.1  DETAILS OF DATASETS

We use four datasets in our evaluation, their details are described below.

**COMPAS**  COMPAS (Larson et al., 2017) dataset contains 5278 data samples. The classification task is to predict recidivism risk from criminal history and demographics. We consider race as the sensitive attribute and include records only with white/black as race. There are 3175 records (60.2%) for the sensitive attribute as white. Among the white group, 52.3% have positive labels while among the black group, this number is 41.9%. Overall, there are 2483 records (47%) are labeled positive.

Table 5: Distribution of data points in clean training dataset and $\mathcal{D}_k$ - Synthetic dataset.

| | Clean | | | $\mathcal{D}_k$ | |
|---|---|---|---|---|---|
| | $y = -$ | $y = +$ | | $y = -$ | $y = +$ |
| $s = 0$ | 18.1% | 36.3% | $s = 0$ | 18.1% | 36.5% |
| $s = 1$ | 31.9% | 13.7% | $s = 1$ | 31.9% | 13.5% |

Table 6: Distribution of data points in clean training dataset and $\mathcal{D}_k$ - MEPS dataset.

| | Clean | | | $\mathcal{D}_k$ | |
|---|---|---|---|---|---|
| | $y = -$ | $y = +$ | | $y = -$ | $y = +$ |
| $s = 0$ | 8.2% | 50.7% | $s = 0$ | 13.0% | 45.7% |
| $s = 1$ | 13.1% | 28.0% | $s = 1$ | 17.6% | 23.7% |

***Pre-processing*** A model trained with the original dataset can only achieve low accuracy (66.6% for the Logistic regression model, compared to the constant prediction classifier that can achieve 53% accuracy), which does not help the understanding of the model's behavior in the presence of adversarial bias. To get rid of the noise that exists in the dataset, we pre-process the dataset as follows: we train an SVM model with RBF kernel on the entire dataset and only keep 60% of the data points which have the smallest loss. To create the training data, test data $\mathcal{D}_{\text{test}}$ and $\mathcal{D}_k$, we randomly split the clean data in the corresponding ratio 4:1:1. Hard examples (the left-out data points) are added to $\mathcal{D}_k$.

***Data distribution*** The data distribution of points in the clean training dataset and $\mathcal{D}_k$ after pre-processing are presented in Table 3. The numbers are the average values over all the datasets we evaluated on. On average, the training data contains 2111 samples, $\mathcal{D}_{\text{test}}$ 528 samples. $\mathcal{D}_k$ consists of 2639 samples out of which 2112 are hard examples. A Logistic regression model trained on the clean data achieves on average 94% accuracy on test data.

**UCI Adult (Census Income)** Adult dataset (Dua & Graff, 2017) includes 48,842 records with 14 attributes such as age, gender, education, marital status, occupation, working hours, and native country. The (binary) classification task is to predict if a person makes over $50K a year based on the census attributes. We consider gender (male and female) as the sensitive attribute. In this dataset, 66.8% are males, and 23.9% are labeled one, i.e having an income over $50K a year. Among male samples, 30.4% are positive samples; for the females, this number is 10.9%.

***Pre-processing*** A model trained on this dataset generally achieves below 90% accuracy (Logistic regression: 85.3%, 2-layer fully connected neural network with 32 hidden units each layer: 85.3% on training data, compared to a constant prediction classifier that can achieve 76.1% accuracy). To enhance the model accuracy, we apply similar pre-processing steps as on COMPAS: we train an SVM model with Linear kernel on the entire dataset and keep 90% of the data points which have the smallest loss. The number of females with income above $50K is small; hence we randomly split 1/2 of the data for $\mathcal{D}_k$. Of the remaining data, 70% are used for training data and 30% for $\mathcal{D}_{\text{test}}$. Hard examples (the left-out data points) are added to $\mathcal{D}_k$.

***Data distribution*** The data distribution of the points in the clean training dataset and $\mathcal{D}_k$ after pre-processing are presented in Table 4. The numbers are the average values over all the datasets we evaluated on. On average, the training data contains 15385 samples, $\mathcal{D}_{\text{test}}$ 6594 samples. $\mathcal{D}_k$ consists of 26863 samples. The training data maintains approximately the same fractions of males and females as in the original dataset. A Logistic regression model trained on the clean data achieves on average 94% accuracy on test data.

**MEPS dataset** Medical Expenditure Panel Survey (MEPS) dataset(mep; ahr) includes 15,675 records with 44 attributes. The classification task is to predict annual hospital utilization (positive label indicates at least 10 visits). We use the same preprocessing as in IBM's AI Fairness 360 (Bellamy et al., 2018) and use race (white or non-white) as the sensitive attribute.

***Pre-processing*** The dataset contains 83% positive samples and 17% negative samples. A logistic regression classifier trained on this data achieves low balanced accuracy (96% for the positive class, and 36% for the negative). Hence, we subsample and keep only 50% of the positive class (for both

groups). We further proceed to train an SVM model with Linear kernel and keep 80% of the data points which have the smallest loss. We randomly split 3/4 of the data for $\mathcal{D}_k$. Of the remaining data, 75% are used for training and 25% for $\mathcal{D}_{\text{test}}$. The left-out data points are added to $\mathcal{D}_k$.

***Data distribution*** The data distribution of points in the clean training dataset and $\mathcal{D}_k$ after preprocessing are presented in Table 6. The numbers are the average values over all the datasets we evaluated on. On average, the training data contains 1372 samples, $\mathcal{D}_{\text{test}}$ 458 samples. $\mathcal{D}_k$ consists of 7321 samples out of which 1381 are hard examples. A Logistic regression model trained on clean data achieves on average 93% accuracy on test data.

**Synthetic data**   As described before, the real-world datasets can be noisy and unbalanced. Hence, we generate a synthetic dataset which does not affect by unbalancing and pre-processing. We used the settings proposed in (Zafar et al., 2019). Specifically, we sample data from two following Guassian distributions $p(x|y = -) = N([2;2],[5,1;1,5])$ and $p(x|y = +) = N([-2;-2],[10,1;1,3])$. For each feature vector $x$, the sensitive attribute is generated by a Bernoulli distribution: $p(s = 1) = p(x'|y = +)/(p(x'|y = +) + p(x'|y = -))$, where $x'$ is a rotated version of $x$. We generate for each label class 1000 training data points, 500 test points and 2000 points for $\mathcal{D}_k$. A Logistic regression model trained on the generated data achieves on average 87.6% accuracy on test data. We do not consider the Hard examples baseline for this dataset.

***Data distribution*** The data distribution of points in the clean training dataset and $\mathcal{D}_k$ after preprocessing are presented in Table 5.

### E.2   IMPLEMENTATION AND PARAMETERS SELECTION

In Algorithm 1 we test with $\lambda \in \{\epsilon, 100\epsilon, 1000\epsilon\}$ and show the results when $\lambda = 1000\epsilon$ for all datasets. In Algorithm 2, we use Logistic regression models. Since we measure the exact $\Delta$ of the model and want that $\Delta$ to have a large impact on finding a new data point in each iteration. We choose $\lambda \in \{\epsilon, 10\epsilon, 100\epsilon\}$ and use $\lambda = 100\epsilon$ in the evaluation for Synthetic, COMPAS and Adult, and $\lambda = \epsilon$ for MEPS. For both algorithms, we use $\eta = 0.001$ as the learning rate.

To train a fair model, we use the post-processing method (Hardt et al., 2016) and reductions approach (Agarwal et al., 2018) and also fair algorithms proposed in (Zhang et al., 2018; Rezaei et al., 2020; Cotter et al., 2019). For post-processing method (Hardt et al., 2016) and reductions approach (Agarwal et al., 2018), we use the implementation of these algorithms provided in (Agarwal et al., 2018)[4]. For the fair algorithm proposed in (Rezaei et al., 2020), we use the implementation provided in (Rezaei et al., 2020)[5]. We use the implementation from AI Fairness 360 library (Bellamy et al., 2018) for the fair algorithm proposed in (Zhang et al., 2018) [6]. We implement the fair algorithm proposed in (Cotter et al., 2019) based on TensorFlow Constrained Optimization library [7].

Note that while the post-processing approach allows achieving exact fairness on the training data, the implementation of the fair algorithms (Cotter et al., 2019; Agarwal et al., 2018) requires a strictly positive $\delta$. We set $\delta = 0$ for the fair algorithm (Cotter et al., 2019). However, the fair algorithms (Rezaei et al., 2020; Zhang et al., 2018; Cotter et al., 2019) does not have a strict fairness constraint that contains $\delta$. We use default values for all hyper-parameters from the available implementation. For the fair algorithm (Zhang et al., 2018), we set the adversary loss weight to 1 which is a hyperparameter that chooses the strength of the adversarial loss. See Eq. (1) in (Zhang et al., 2018). It is important to note that, the output of reduction, post-processing approaches, and fair algorithm (Cotter et al., 2019) is a *randomized classifier*. We, therefore, use the expected accuracy to measure the classification performance, given by

$$\text{Acc}(\theta; \mathcal{D}) = 1 - \frac{1}{|\mathcal{D}|} \sum_{(x,y) \in \mathcal{D}} |f_\theta(x) - y|, \tag{9}$$

where $f_\theta(x)$ is the expected prediction of randomized classifier $f_\theta$. For the unconstrained models, $f_\theta$ is the deterministic prediction.

---

[4] https://github.com/fairlearn/fairlearn
[5] https://github.com/arezae4/fair-logloss-classification
[6] https://github.com/Trusted-AI/AIF360
[7] https://github.com/google-research/tensorflow_constrained_optimization

### E.3 ROBUSTNESS EVALUATION

In this section, we provide the detailed results about the test accuracy and fairness gap of the target models for all datasets, as discussed in Section 5. We show that the fair models are more vulnerable to adversarial bias than the unconstrained model. In addition, the test accuracy and the fairness property of fair models are both compromised.

**Test accuracy**    In the Table 7, we compare the effect of adversarial bias on unconstrained models (without fairness constraint) with fair models trained with different fair algorithms on all datasets when the amount of bias equals 10% of the training data i.e., $\epsilon = 0.1$. On all the datasets, we observe a larger relative accuracy drop for all fair models compared with unconstrained models. This implies that, under the adversarial bias setting, *the fairness constraints hurt the robustness of the model*.

**Fairness gap**    In Table 8, we compare the effect of adversarial bias on unconstrained model (without fairness constraint) with different fair models at different $\delta$ on all datasets, when $\epsilon = 0.1$. We notice that the fairness gap of fair models trained on the biased data is larger than those of fair models trained on clean data (as shown in "Benign" row) (this observation is particularly pronounced on COMPAS, Adult, and Synthetic datasets). This implies fair models trained on biased data become less fair on the test dataset when biases are present in both adversarial sampling bias and adversarial labeling bias setting. This shows that not only does adversarial bias can cause accuracy drops, but it is also able to make fair models more discriminatory on test data.

### E.4 CONFLICT BETWEEN FAIRNESS AND ROBUSTNESS

In Figure 1, Figure 7, Figure 8 and Figure 9 we compare the test accuracy of target model at different fractions of adversarial bias selected using all the strategies on COMPAS, Adult, Synthetic and MEPS datasets respectively. The x-axis $\epsilon$ is the ratio of the size of $\mathcal{D}_p$ to the size of the clean dataset and reflects the contamination level of the training set. We compare the impact of adversarial bias with baselines and adversarial bias against unconstrained models, for various $\epsilon$. The difference between test accuracy at $\epsilon = 0$ (benign setting) and a larger $\epsilon$ values reflects the impact of the biasing strategy. Constant prediction always outputs the majority label in the clean dataset.

For the Adult dataset, notice that the Constant prediction baseline has good accuracy (>80%). Hence, the relative accuracy drop on the Adult dataset is not as significant as that on the COMPAS dataset. However, we can still observe similar results that compared to the unconstrained models, the fair models suffer a greater accuracy drop, with our proposed algorithms perform significantly better than the baselines. The three algorithms have similar results both when the fair models are trained with (Hardt et al., 2016) and (Agarwal et al., 2018).

For the Synthetic and MEPS datasets, adversarial sampling hardly influences the unconstrained models, while some small effect is observed on the fair models. On the other hand, on Synthetic data, adversarial labeling bias has a much stronger effect on the fair models than on the unconstrained models. For MEPS, the difference in accuracy is smaller; yet we can still observe the accuracy drop is larger for the fair models. This is another evidence for the claim that fair models are less robust to bias than unconstrained models.

### E.5 EFFECT OF FAIRNESS LEVEL ON IMPACT OF ADVERSARIAL BIAS

In Figure 10, Figure 11, Figure 12 and Figure 13, we show the effect of fairness level $\delta$ on impact of adversarial sampling bias and adversarial labeling bias for COMPAS, Adult, Synthetic and MEPS dataset respectively. To measure the influence of fairness level $\delta$, we generate bias using Algorithm 2 and Algorithm 1 for both adversarial labeling and adversarial sampling settings.

We measure the test accuracy of models learned with different values of fairness level $\delta$ on the same biased dataset. We can observe that the drop in accuracy for the same fraction of bias is higher for models with stricter fairness constraints (smaller $\delta$). This shows that the more fair a model tries to be, the more vulnerable it becomes to adversarial bias. We also present the majority (the protected group with a larger number of samples) accuracy and minority accuracy. It is clear that the accuracy drop for the majority is more significant than that for minorities for all the cases.

In the Appendix E.10, we show that the algorithms choose the points with large loss from the smallest subgroup (subgroups are determined by the protected attribute and the label). As a result, in order to achieve fairness on the biased dataset, fair models are more likely to reduce the accuracy of the majority group.

On Synthetic dataset, this observation is not clear as the size of the majority is not much bigger than that of the minority (54.4% vs. 45.6%). On MEPS dataset, the accuracy drop for the majority and minority is similar. From Figure 26, we notice that $\mathcal{D}_\mathrm{p}$ generated by Algorithm 2 with $\lambda = \epsilon$ and Algorithm 2 with $\lambda = \epsilon$ has a similar fraction of samples from the each group. This is because the fairness gap of the unconstrained model on MEPS is small. In addition, the selected $\lambda$ for MEPS is relatively small. As a result, data points that have a large loss also have a large weighted sum of the loss and the fairness gap. As a consequence, $D_\mathrm{p}$ generated by Algorithm 2 with $\lambda = 0$ and Alghorithm 2 with $\lambda = \epsilon$ has a similar effect on the majority and minority of the fair models.

## E.6    TRANSFERABILITY OF ADVERSARIAL BIAS.

Table 9 shows the test accuracy of fair models when we generate poisoning data without the knowledge of the exact data points in the clean training data $\mathcal{D}_\mathrm{c}$ or the target model's architecture. Instead, we use a substitute model and a substitute dataset drawn from the same distribution. The target model's architecture used in the unconstrained model and the fair models trained with (Agarwal et al., 2018) and (Hardt et al., 2016) is the decision tree. However, to generate poisoning data, we use a logistic regression model instead of a decision tree in Algorithm 1 and Algorithm 2. We observe that for both adversarial sampling and adversarial labeling, there is a similar pattern in the model performance and impact of the bias compared to the results in Table 1, where we used the knowledge of the clean training data to generate the poisoning data. This implies that the adversarial bias can still be effectively transferred to the more realistic scenarios where knowledge of the target's model and clean training data is not available to an adversary. It means, in practice, an adversary who only knows the fairness notion that the target model aims to satisfy can successfully reduce the overall accuracy of the resulting model significantly.

## E.7    OTHER NOTIONS OF GROUP FAIRNESS

We present experimental results for equal opportunity to show the vulnerability of fair models under other notions of group fairness. Without loss of generality, we consider positive prediction as the advantaged outcome. Equal opportunity, a relaxation of equalized odds, enforces non-discrimination among groups with "advantaged" prediction, i.e, it requires the true positive rate to be equal among all groups. We use a relaxed notion of equal opportunity, formally defined as follows:

**Definition 1** (Equal opportunity). *A binary classifier $f_\theta$ is $\delta$-fair under equal opportunity if*

$$\Delta_{\mathrm{EOpp}}(\theta, \mathcal{D}) \triangleq \left| \Pr_{\mathcal{D}}[f_\theta(X) \neq y | S = 0, Y = +] - \Pr_{\mathcal{D}}[f_\theta(X) \neq y | S = 1, Y = +] \right| \leq \delta, \quad (10)$$

*where, the probabilities are computed empirically over the training set $\mathcal{D}$. We refer to $\Delta_{\mathrm{EOpp}}$ as the model's empirical fairness gap under equal opportunity. A model satisfies exact fairness under equal opportunity when $\delta = 0$.*

Algorithm 1 and Algorithm 2 can be modified accordingly for equal opportunity by replacing $\Delta$ with $\Delta_{\mathrm{EOpp}}$. In Figure 14, we compare the test accuracy of unconstrained model and fair models with respect to equal opportunity when $\epsilon$ varies for the reduction approach (Fair (Agarwal et al., 2018), $\delta = 0.01$) and post-processing approach (Fair (Hardt et al., 2016), $\delta = 0$) on COMPAS dataset. We observe the same pattern as in Figure 1 that at the same $\epsilon$, the accuracy drop of the fair models is significantly more than that of the unconstrained models. We can come to the conclusion that fair models trained with equal opportunity are noticeably less robust than unconstrained models. In addition, comparing the unconstrained model with fair models, as $\epsilon$ increases, shows that the cost of fairness increases. Hence, for equal opportunity fairness notion, adversarial bias amplifies the cost of fairness.

## E.8    ACCURACY ON TRAINING DATA FOR MINORITY GROUP

In Figure 15, Figure 16, Figure 17 and Figure 18, the accuracy of the unconstrained model on $\mathcal{D}_\mathrm{p}$ from the minority group is compared with the corresponding accuracy of fair models with different

fairness level $\delta$ on COMPAS, Adult, Synthetic and MEPS datasets. All fair models are trained with the reductions approach (Agarwal et al., 2018). $\mathcal{D}_\mathrm{p}$ is selected using Algorithm 2 with $\lambda = 0$ for the unconstrained model for both adversarial labeling and adversarial sampling settings. On all datasets, we can observe that, for the fair model using (Agarwal et al., 2018), as the value of $\delta$ decreases, the accuracy of the model increases on $\mathcal{D}_\mathrm{p}$ and decreases on clean training data. This implies adversarial bias reduces fair models' ability to learn from clean data on the minority group.

### E.9 Fairness Gap of the Unconstrained Model on Biased Training Data

To investigate the effect of adversarial bias, we train an unconstrained classifier without any fairness constraints and measure the fairness gap $\Delta(\theta; \mathcal{D}_\mathrm{c} \cup \mathcal{D}_\mathrm{p})$ of the poisoned training dataset generated by different algorithms. Figure 19, Figure 20, Figure 21 and Figure22 show the results for all datasets. We observe a correlation between the effect of the bias and the corresponding fairness gap on the training data. For the baselines (Label flipping, Random sampling, Hard examples), the slight increase in $\Delta$ corresponds to a small accuracy drop on the test data. For our biasing strategies, $\Delta$ is much larger and at the same time, the corresponding test accuracy is also significantly lower than observed for the baselines.

### E.10 Distribution of $\mathcal{D}_\mathrm{p}$

In Figure 23, Figure 24, Figure 25, and Figure 26 we show group membership based on the protected attribute and labels of the data which are generated via different biasing strategies. The number of samples with $s = 1, y = +$ ($s = 0, y = -$ for MEPS) is the smallest among the four combinations of labels and the protected attribute. As shown in each figure, the biasing algorithms in the first two rows are more effective compared with baselines in the second row. As shown in sub-figures (a)-(f), in more effective strategies, most data points in $\mathcal{D}_\mathrm{p}$ are from the smallest subgroup (positive labeled points from the minority).

### E.11 Performance of Algorithm 2 with $\lambda = 100\epsilon$ on Adult dataset

We notice that there are accuracy fluctuations for the fair models evaluated on the $\mathcal{D}_\mathrm{p}$ selected by Algorithm 2 with $\lambda = 100\epsilon$ on the Adult dataset. Recall that, the algorithm selects data from the $\mathcal{D}_\mathrm{k}$ *without replacement*. In each iteration, it selects the data point that maximizes the classification loss plus the fairness gap (as at Line 5 in Algorithm 2). As shown in Figure 23, Figure 24 and Figure 25, Algorithm 2 with $\lambda = 100\epsilon$ has a significant preference to select data that would result in a large fairness gap. Thus, it chooses data that would fall into the smallest subgroup in the training set. This is shown to be very effective in the case of COMPAS dataset and can lead to a sharp decrease in the model accuracy even for small $\epsilon$ (see Figure 1). However, this greedy algorithm in the case of small $\mathcal{D}_\mathrm{k}$, and no repetition in the $\mathcal{D}_\mathrm{p}$, can result in the degradation of the impact of the adversarial bias for larger $\epsilon$ values, as we see in Figure 7.

In more detail, the reason behind the Algorithm 2 with $\lambda = 100\epsilon$ for larger $\epsilon$ on the Adult dataset is the following. In the adversarial sampling setting, the size of the smallest subgroup ($y = +$ and $s = 1$) in the $\mathcal{D}_\mathrm{k}$ is only equivalent to $\epsilon = 7.69\%$ of $\mathcal{D}_\mathrm{p}$. For larger values of $\epsilon$, Algorithm 2 with $\lambda = 100\epsilon$ will choose data from other subgroups, which cannot further harm the model accuracy, thus reduces the effect of the adversarial bias.

In the adversarial labeling setting, with large $\epsilon$, the number of $\mathcal{D}_\mathrm{p}$ is larger than the size of subgroups with positive labels ($y = +$) in $\mathcal{D}_\mathrm{c}$; typically when $\epsilon = 0.2$, $|\mathcal{D}_\mathrm{p}| > 3000$ whereas the number of samples with $y = +$ in $\mathcal{D}_\mathrm{c}$ is 2943 on average. Relying on choosing points to select data points to maximize $\Delta$ results in the possibility of choosing points from any subgroups with positive labels (as shown in Figure 16(b)), since data points can dominate any of these subgroups.

In summary, the fluctuation in the figures is due to the significant effect of maximizing the fairness gap. In fact, in both adversarial sampling and labeling settings, Algorithm 2 ($\lambda = 100\epsilon$) achieves the same performances with a smaller $\epsilon$ as the other algorithms with larger $\epsilon$. These results, in effect, reflect the effectiveness of the algorithm.

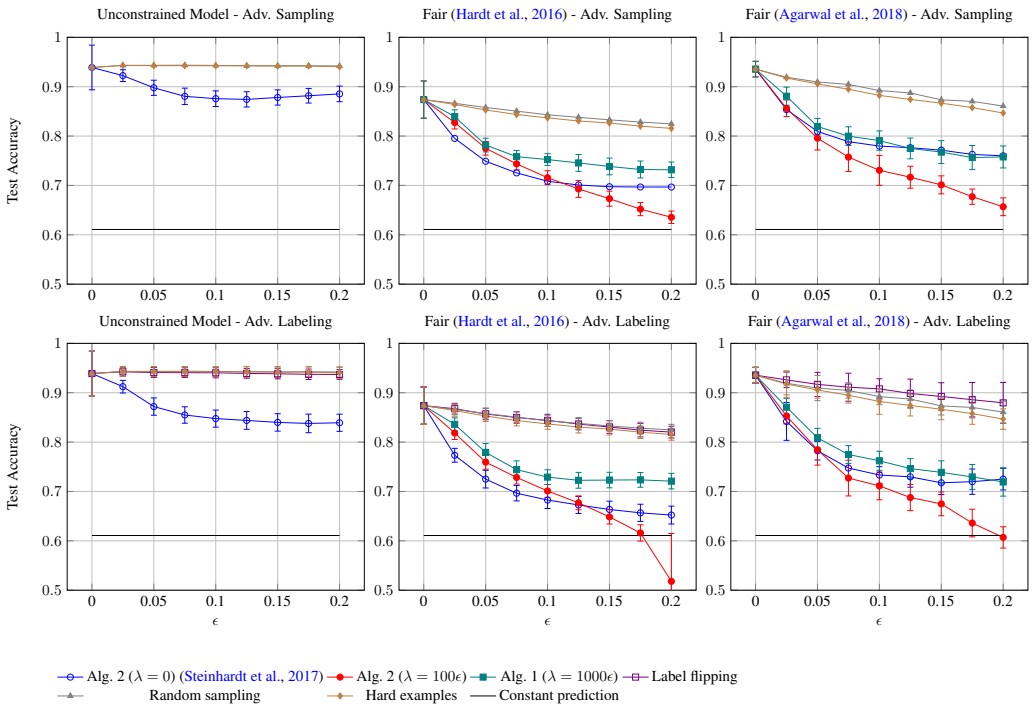

Figure 6: Test accuracy of unconstrained and fair models under adversarial bias – COMPAS dataset. The x-axis $\epsilon$ is the ratio of the size of biasing dataset $\mathcal{D}_{\mathrm{p}}$ to the size of clean dataset, and reflects the contamination level of training set. We compare the impact of adversarial bias with baselines and poisoning attacks against unconstrained models, for various $\epsilon$. The difference between test accuracy at $\epsilon = 0$ (benign setting) and larger $\epsilon$ values reflects the impact of the biasing strategy. Constant prediction always outputs the majority label in clean dataset. The enforced fairness level $\delta$ is 0 and 0.01 for the fair model (Hardt et al., 2016) and fair model (Agarwal et al., 2018), respectively.

Table 7: Test accuracy of unconstrained models and fair models (Hardt et al., 2016; Agarwal et al., 2018; Rezaei et al., 2020; Cotter et al., 2019; Zhang et al., 2018) in the presence of adversarial bias - all datasets, for $\epsilon = 0.1$. We report and compare the accuracy of the unconstrained models and fair models in the benign ($\epsilon = 0$) and adversarial bias setting. When $\epsilon = 0$, the models are trained on $\mathcal{D}_c$. For each result in the adversarial bias setting, we report the relative accuracy drop compared with the accuracy in the benign setting ($\epsilon = 0$). The smaller relative accuracy drop reflects the stronger robustness of the model. The numbers in bold are the *largest* relative accuracies drop of the target models against different algorithms.

| Dataset | Target Model | $\epsilon = 0$ | Adversarial Sampling ($\epsilon = 0.1$) | | | Adversarial Labeling ($\epsilon = 0.1$) | | |
|---|---|---|---|---|---|---|---|---|
| | | | Alg. 2 ($\lambda = 0$) | Alg. 2 | Alg. 1 | Alg. 2 ($\lambda = 0$) | Alg. 2 | Alg. 1 |
| Synthetic | Unconstrained | 87.6 | 87.4 (**0.2↓**) | - | - | 85.1 (**2.9%↓**) | - | - |
| | Fair (Hardt et al., 2016) | 80.0 | 77.2 (**3.5%↓**) | 78.9 (1.4%↓) | 79.9 (0.1%↓) | 71.0 (11.%↓) | 71.1 (11.%↓) | 70.9 (**11.%↓**) |
| | Fair (Agarwal et al., 2018) | 84.9 | 80.9 (**4.7%↓**) | 82.8 (2.5%↓) | 81.7 (3.8%↓) | 75.2 (11.%↓) | 68.0 (**20.%↓**) | 68.0 (20.%↓) |
| | Fair (Rezaei et al., 2020) | 76.1 | 70.2 (**7.8%↓**) | 71.4 (6.2%↓) | 74.2 (2.5%↓) | 60.9 (20.%↓) | 58.3 (**23.%↓**) | 60.1 (21.%↓) |
| | Fair (Cotter et al., 2019)) | 85.5 | 84.5 (1.2%↓) | 84.5 (**1.2%↓**) | 84.9 (0.7%↓) | 80.3 (6.1%↓) | 79.7 (6.8%↓) | 77.8 (**9.0%↓**) |
| | Fair (Zhang et al., 2018) | 78.1 | 75.0 (**4.0%↓**) | 75.1 (3.8%↓) | 76.4 (2.2%↓) | 66.1 (15.%↓) | 63.0 (**19.%↓**) | 65.3 (16.%↓) |
| COMPAS | Unconstrained | 94.3 | 87.6 (**7.1%↓**) | - | - | 84.7 (**10.%↓**) | - | - |
| | Fair (Hardt et al., 2016) | 87.4 | 70.8 (**19.%↓**) | 71.6 (18.%↓) | 75.2 (14.%↓) | 68.2 (**22.%↓**) | 69.8 (20.%↓) | 72.9 (17.%↓) |
| | Fair (Agarwal et al., 2018) | 93.6 | 77.8 (17.%↓) | 73.1 (**22.%↓**) | 79.0(16.%↓) | 73.0 (22.%↓) | 67.7 (**28.%↓**) | 76.2 (19.%↓) |
| | Fair (Rezaei et al., 2020) | 84.5 | 63.2 (**25.%↓**) | 64.7 (23.%↓) | 68.2 (19.%↓) | 60.9 (28.%↓) | 60.6 (**28.%↓**) | 66.2 (22.%↓) |
| | Fair (Cotter et al., 2019) | 91.9 | 79.6 (**13.%↓**) | 81.2 (12.%↓) | 82.9 (10.%↓) | 76.2 (**17.%↓**) | 76.7 (17.%↓) | 79.3 (14.%↓) |
| | Fair (Zhang et al., 2018) | 81.0 | 70.0 (14.%↓) | 66.0 (**19.%↓**) | 72.3 (11.%↓) | 67.4 (17.%↓) | 64.1 (**21.%↓**) | 70.9 (12.%↓) |
| Adult | Unconstrained | 94.3 | 94.0 (**0.3%↓**) | - | - | 89.3 (**5.3%↓**) | - | - |
| | Fair (Hardt et al., 2016) | 92.7 | 89.6 (3.3%↓) | 90.1 (2.8%↓) | 91.4 (1.4%↓) | 84.6 (8.7%↓) | 81.1 (**13.%↓**) | 84.8 (8.5%↓) |
| | Fair (Agarwal et al., 2018) | 93.8 | 91.7 (2.2%↓) | 92.2 (1.7%↓) | 92.4 (1.5%↓) | 83.9 (11.%↓) | 80.9 (**14.%↓**) | 86.1 (8.2%↓) |
| | Fair (Rezaei et al., 2020) | 84.5 | 78.1 (7.6%↓) | 79.1 (6.4%↓) | 78.9 (6.6%↓) | 70.4 (17.%↓) | 64.9 (**23.%↓**) | 76.1 (9.9%↓) |
| | Fair (Cotter et al., 2019) | 92.7 | 91.2 (1.6%↓) | 91.1 (1.7%↓) | 90.3 (**2.6%↓**) | 84.0 (9.1%↓) | 83.8 (**9.6%↓**) | 86.8 (6.4%↓) |
| | Fair (Zhang et al., 2018) | 89.7 | 85.7 (**4.4%↓**) | 86.1 (4.0%↓) | 88.7 (1.1%↓) | 82.6 (7.9%↓) | 44.9 (**50.%↓**) | 51.9 (42.%↓) |
| MEPS | Unconstrained | 92.8 | 91.2 (**1.7%↓**) | - | - | 88.8 (**4.3%↓**) | - | - |
| | Fair (Hardt et al., 2016) | 91.3 | 88.6 (3.0%↓) | 87.8 (**3.8%↓**) | 90.4 (1.0%↓) | 85.0 (**6.9%↓**) | 85.2 (6.7%↓) | 90.1 (1.3%↓) |
| | Fair (Agarwal et al., 2018) | 92.6 | 89.8 (3.0%↓) | 88.9 (**4.0%↓**) | 91.1 (1.6%↓) | 85.9 (**7.2%↓**) | 86.0 (7.1%↓) | 90.7 (2.0%↓) |
| | Fair (Rezaei et al., 2020) | 88.6 | 75.7 (**15.%↓**) | 76.0 (14.%↓) | 86.5 (2.4%↓) | 74.8 (16.%↓) | 74.5 (**16.%↓**) | 86.0 (2.9%↓) |
| | Fair (Cotter et al., 2019) | 91.4 | 89.1 (2.5%↓) | 88.2 (**3.5%↓**) | 89.9 (1.6%↓) | 86.9 (4.9%↓) | 86.8 (**5.0%↓**) | 89.4 (2.2%↓) |
| | Fair (Zhang et al., 2018) | 78.6 | 67.2 (**15.%↓**) | 66.8 (15.%↓) | 75.9 (3.4%↓) | 63.7 (**19.%↓**) | 64.4 (18.%↓) | 74.2 (5.6%↓) |

Table 8: **Fairness gap of target models on test data** for $\epsilon = 0.1$. Fair models are trained using the reduction approach (Agarwal et al., 2018). The fairness gap $\Delta$ is defined in (1). The numbers reflect how unfair the model is with respect to the protected group in the test data. For fair models, compare numbers with $\delta$ (the guaranteed fairness gap on training data). The farther apart $\Delta$ and $\delta$ are, the less the fairness generalization is on test data.

| Dataset | Strategies | Unconstrained Model | Fair $(\delta = 0.1)$ | Fair $(\delta = 0.05)$ | Fair $(\delta = 0.01)$ |
|---|---|---|---|---|---|
| Synthetic | Benign | 0.18±0.04 | 0.14±0.05 | 0.08±0.04 | 0.03±0.02 |
| | Random Sampling | 0.18±0.04 | 0.14±0.04 | 0.08±0.04 | 0.03±0.02 |
| | Label flipping | 0.19±0.04 | 0.07±0.04 | 0.07±0.02 | 0.10±0.03 |
| | Adv. sampling (Alg. 2, $\lambda = 0$) | 0.17±0.05 | 0.08±0.05 | 0.08±0.04 | 0.07±0.04 |
| | Adv. sampling (Alg. 1, $\lambda = 1000\epsilon$) | - | 0.05±0.03 | 0.04±0.03 | 0.04±0.02 |
| | Adv. sampling (Alg. 2, $\lambda = 100\epsilon$) | - | 0.07±0.04 | 0.09±0.04 | 0.09±0.04 |
| | Adv. labeling (Alg. 2, $\lambda = 0$) | 0.29±0.10 | 0.07±0.04 | 0.10±0.04 | 0.11±0.06 |
| | Adv. labeling (Alg. 1, $\lambda = 1000\epsilon$) | - | 0.08±0.03 | 0.15±0.03 | 0.14±0.06 |
| | Adv. labeling (Alg. 2, $\lambda = 100\epsilon$) | | 0.12±0.05 | 0.15±0.05 | 0.14±0.07 |
| COMPAS | Benign | 0.21±0.07 | 0.11±0.06 | 0.08±0.05 | 0.06±0.04 |
| | Random Sampling | 0.19±0.07 | 0.08±0.03 | 0.10±0.03 | 0.11±0.05 |
| | Hard examples | 0.19±0.08 | 0.09±0.03 | 0.11±0.03 | 0.13±0.05 |
| | Label flipping | 0.23±0.07 | 0.09±0.04 | 0.08±0.03 | 0.07±0.04 |
| | Adv. sampling (Alg. 2, $\lambda = 0$) | 0.26±0.08 | 0.19±0.07 | 0.25±0.07 | 0.30±0.07 |
| | Adv. sampling (Alg. 1, $\lambda = 1000\epsilon$) | - | 0.16±0.05 | 0.20±0.07 | 0.22±0.09 |
| | Adv. sampling (Alg. 2, $\lambda = 100\epsilon$) | - | 0.29±0.06 | 0.33±0.08 | 0.37±0.09 |
| | Adv. labeling (Alg. 2, $\lambda = 0$) | 0.28±0.08 | 0.13±0.05 | 0.15±0.07 | 0.19±0.08 |
| | Adv. labeling (Alg. 1, $\lambda = 1000\epsilon$) | - | 0.14±0.05 | 0.17±0.07 | 0.16±0.08 |
| | Adv. labeling (Alg. 2, $\lambda = 100\epsilon$) | - | 0.28±0.05 | 0.31±0.09 | 0.39±0.08 |
| Adult | Benign | 0.07±0.03 | 0.07±0.03 | 0.06±0.03 | 0.04±0.02 |
| | Random Sampling | 0.07±0.03 | 0.07±0.03 | 0.06±0.03 | 0.03±0.02 |
| | Hard examples | 0.08±0.03 | 0.06±0.03 | 0.06±0.03 | 0.04±0.02 |
| | Label flipping | 0.08±0.04 | 0.10±0.04 | 0.17±0.04 | 0.24±0.04 |
| | Adv. sampling (Alg. 2, $\lambda = 0$) | 0.06±0.03 | 0.03±0.02 | 0.07±0.03 | 0.12±0.03 |
| | Adv. sampling (Alg. 1, $\lambda = 1000\epsilon$) | - | 0.03±0.02 | 0.05±0.02 | 0.09±0.03 |
| | Adv. sampling (Alg. 2, $\lambda = 100\epsilon$) | - | 0.06±0.03 | 0.07±0.02 | 0.07±0.02 |
| | Adv. labeling (Alg. 2, $\lambda = 0$) | 0.06±0.04 | 0.07±0.03 | 0.13±0.04 | 0.21±0.04 |
| | Adv. labeling (Alg. 1, $\lambda = 1000\epsilon$) | - | 0.1±0.04 | 0.16±0.04 | 0.13±0.05 |
| | Adv. labeling (Alg. 2, $\lambda = 100\epsilon$) | - | 0.18±0.06 | 0.16±0.06 | 0.09±0.08 |
| MEPS | Benign | 0.09±0.06 | 0.08±0.05 | 0.08±0.05 | 0.08±0.05 |
| | Random Sampling | 0.09±0.06 | 0.09±0.07 | 0.07±0.05 | 0.08±0.06 |
| | Hard examples | 0.11±0.06 | 0.08±0.05 | 0.07±0.04 | 0.08±0.05 |
| | Label flipping | 0.08±0.05 | 0.07±0.05 | 0.07±0.05 | 0.07±0.05 |
| | Adv. sampling (Alg. 2, $\lambda = 0$) | 0.09±0.05 | 0.10±0.06 | 0.20±0.10 | 0.14±0.08 |
| | Adv. sampling (Alg. 1, $\lambda = 1000\epsilon$) | - | 0.08±0.05 | 0.09±0.06 | 0.08±0.05 |
| | Adv. sampling (Alg. 2, $\lambda = \epsilon$) | - | 0.09±0.06 | 0.20±0.10 | 0.13±0.08 |
| | Adv. labeling (Alg. 2, $\lambda = 0$) | 0.08±0.07 | 0.09±0.07 | 0.10±0.07 | 0.09±0.07 |
| | Adv. labeling (Alg. 1, $\lambda = 1000\epsilon$) | - | 0.07±0.04 | 0.07±0.05 | 0.07±0.04 |
| | Adv. labeling (Alg. 2, $\lambda = \epsilon$) | - | 0.11±0.08 | 0.10±0.07 | 0.09±0.07 |

Table 9: Test accuracy of unconstrained models and fair models in the presence of adversarial bias – COMPAS for $\epsilon = 0.1$. We report and compare the accuracy of the unconstrained models and fair models in adversarial bias setting when the clean training data and the target model's architecture are unknown. Instead, a substitute dataset and a substitute model are used in Algorithm 1 and Algorithm 2. We also report the relative accuracy drop in parenthesis.

| Target Model | Adv. Sampling ($\epsilon = 0.1$) | Adv. Labeling ($\epsilon = 0.1$) |
|---|---|---|
| Unconstrained | 92.2 (4.6%↓) | 91.7 (5.1%↓) |
| Fair (Hardt et al., 2016) ($\delta = 0$) | 83.6 (13.%↓) | 82.6 (14.%↓) |
| Fair (Agarwal et al., 2018) ($\delta = 0.01$) | 84.1 (13.%↓) | 82.0 (15.%↓) |

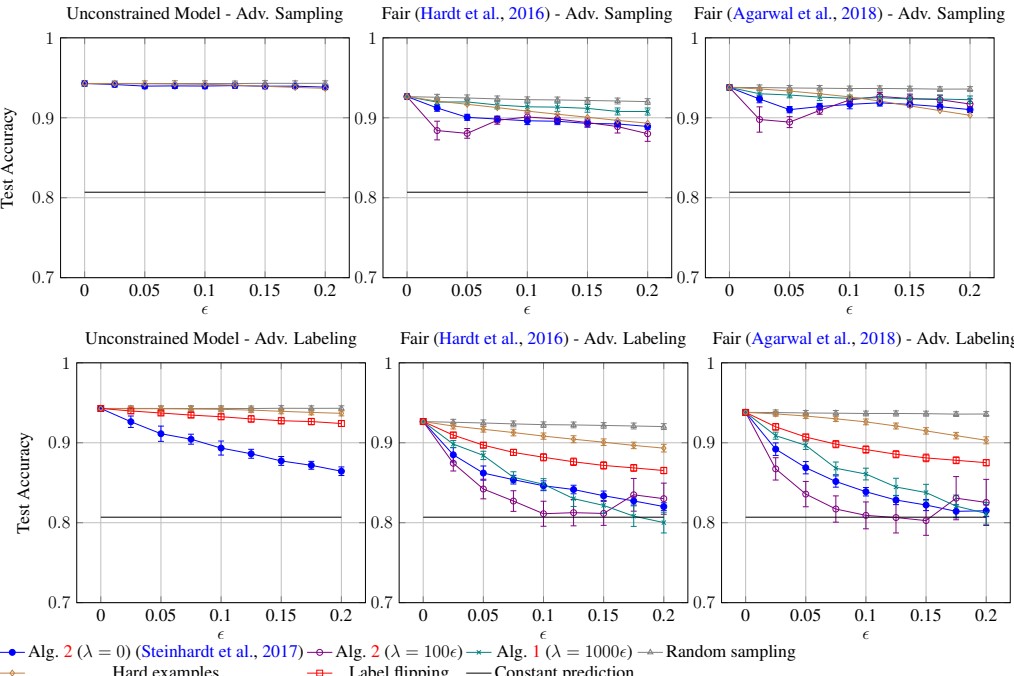

Figure 7: Test accuracy of unconstrained and fair models under adversarial bias – Adult dataset. The enforced fairness level $\delta$ is 0 and 0.01 for the fair model (Hardt et al., 2016) and fair model (Agarwal et al., 2018), respectively.

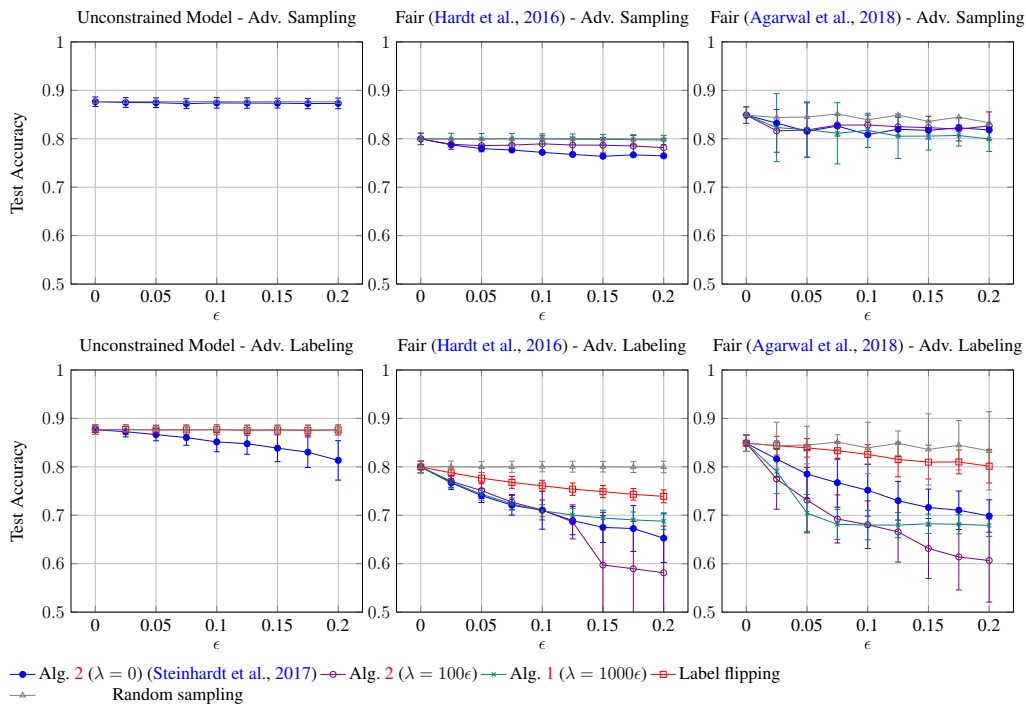

Figure 8: Test accuracy of unconstrained and fair models under adversarial bias – Synthetic dataset. Constant prediction has 50% accuracy. The enforced fairness level $\delta$ is 0 and 0.01 for the fair model (Hardt et al., 2016) and fair model (Agarwal et al., 2018), respectively.

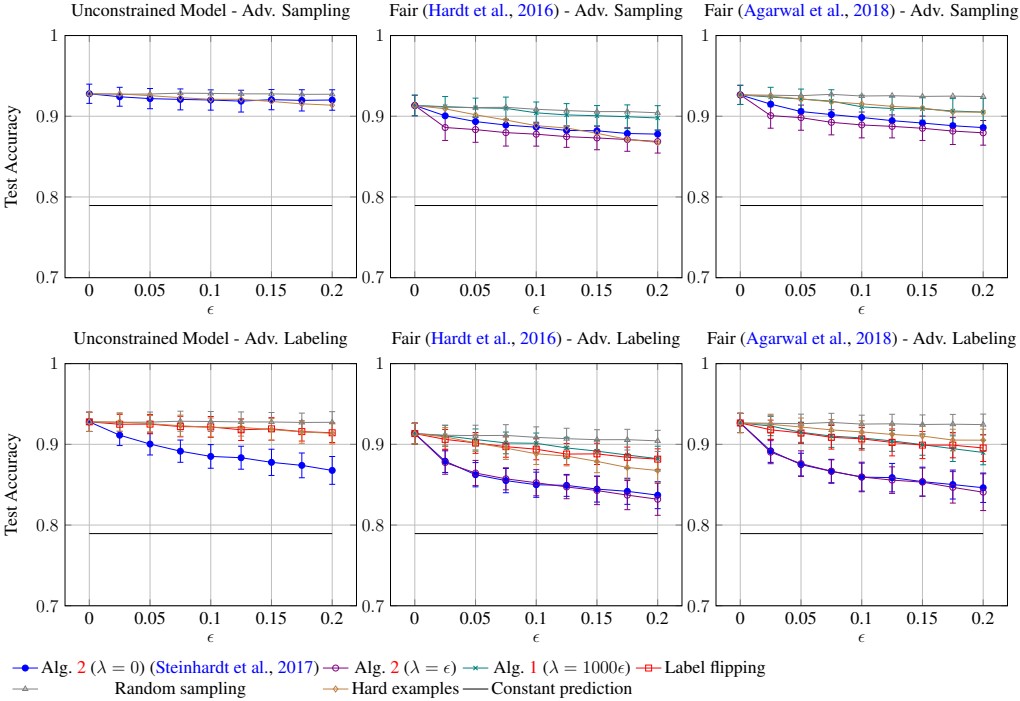

Figure 9: Test accuracy of unconstrained and fair models under adversarial bias – MEPS dataset. The enforced fairness level $\delta$ is 0 and 0.01 for the fair model (Hardt et al., 2016) and fair model (Agarwal et al., 2018), respectively.

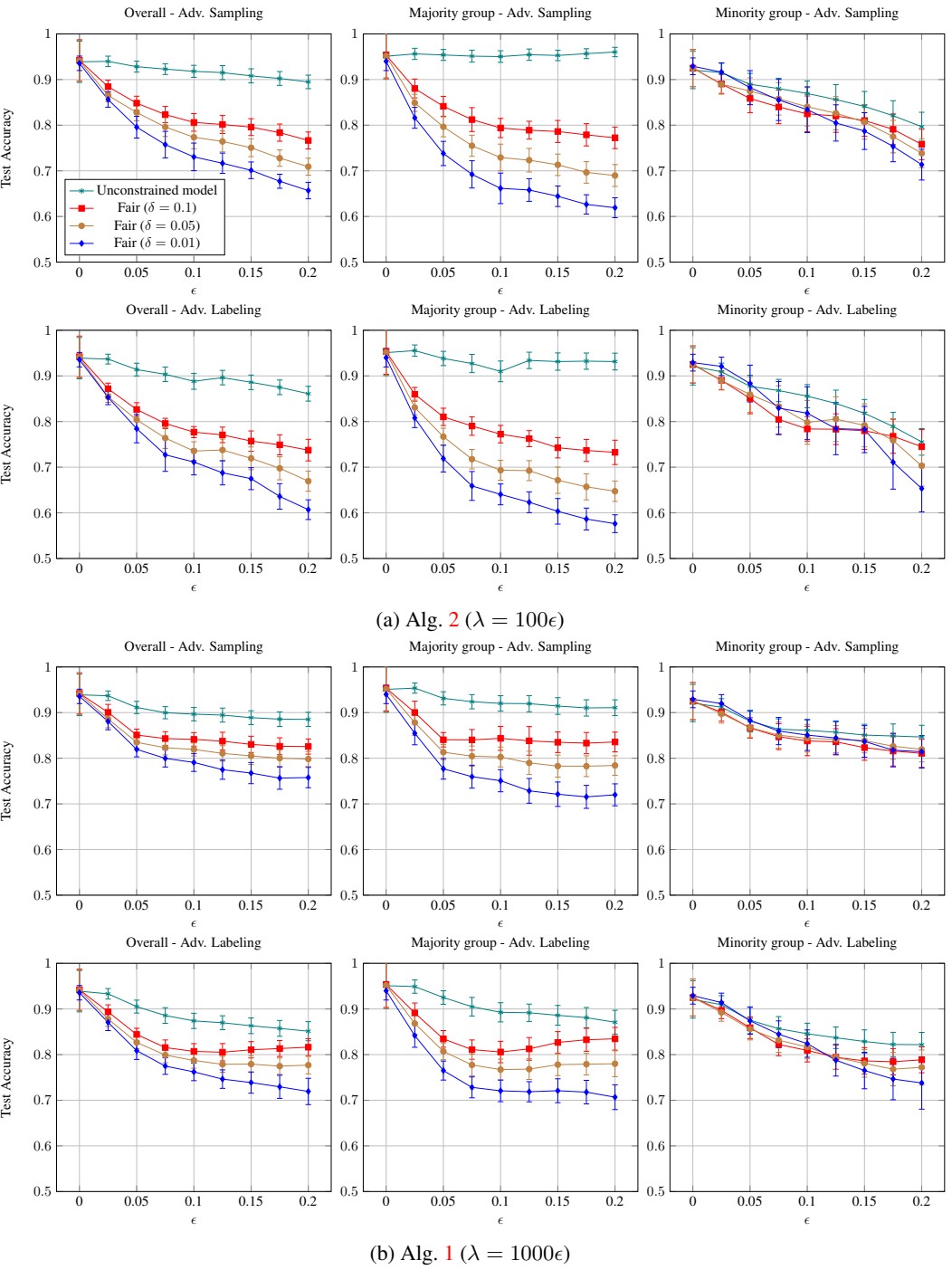

Figure 10: Effect of fairness level $\delta$ on robustness across groups under adversarial sampling and adversarial labeling bias – COMPAS dataset. The reductions approach (Agarwal et al., 2018) is used to train fair models. The majority group (whites) contributes 61% of the test data.

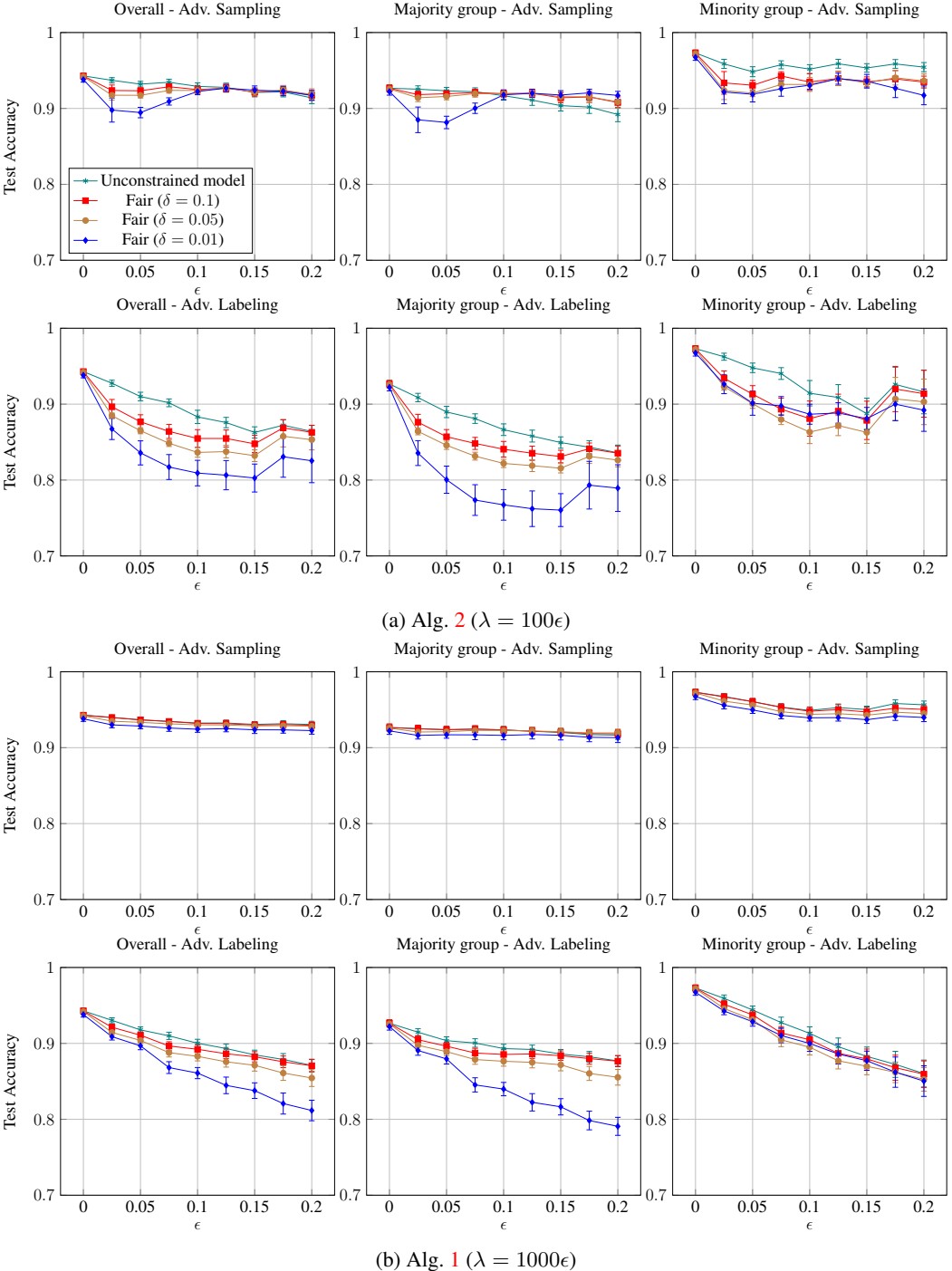

Figure 11: Effect of fairness level $\delta$ on robustness across groups under adversarial sampling and adversarial labeling bias – Adult dataset. The majority group (males) contributes 66% of the test data. The reductions approach (Agarwal et al., 2018) is used to train fair models.

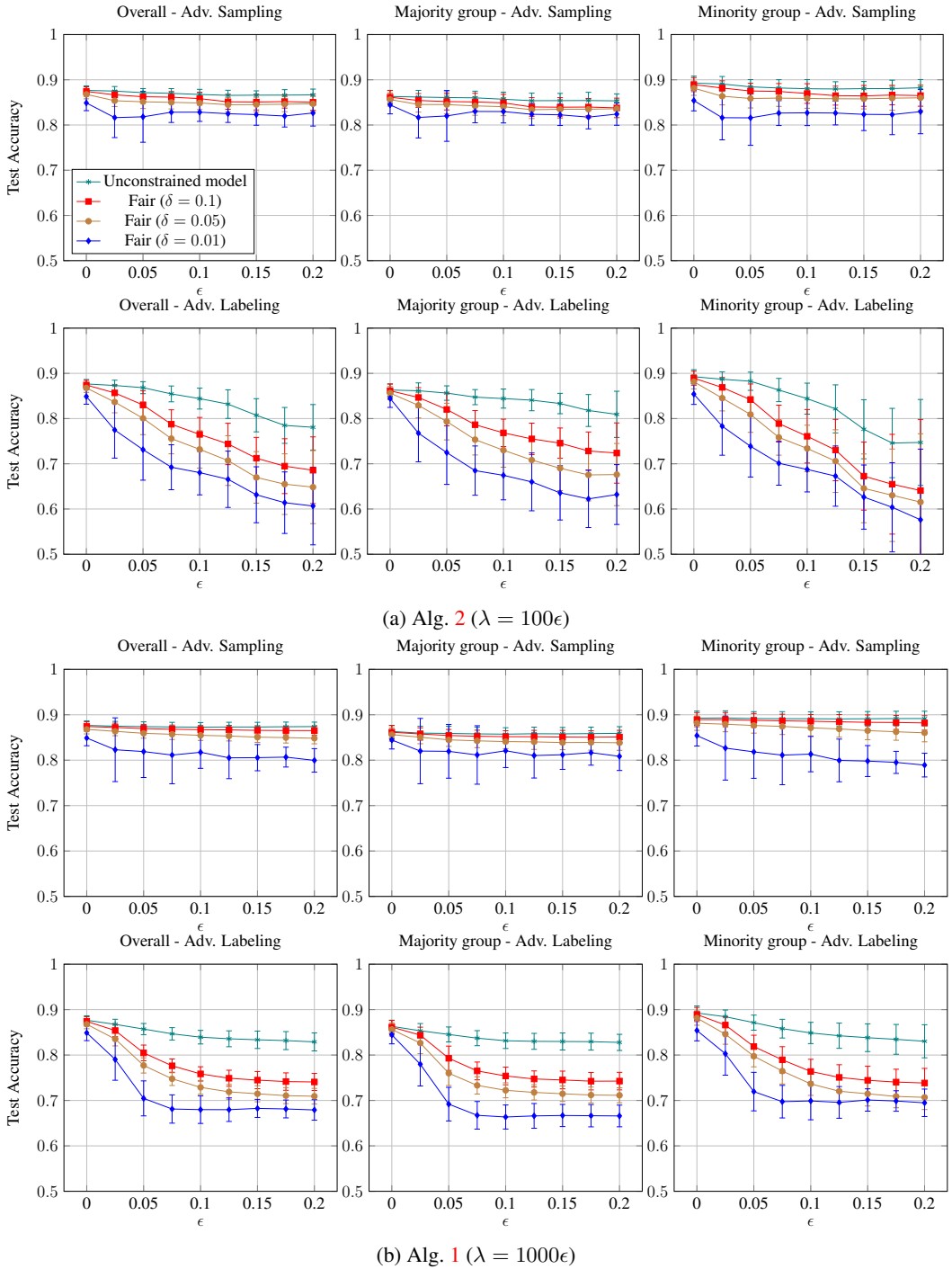

Figure 12: Effect of fairness level $\delta$ on robustness across groups under adversarial sampling and adversarial labeling bias – Synthetic dataset. The majority group contributes 54.4% of the test data. The reductions approach (Agarwal et al., 2018) is used to train fair models.

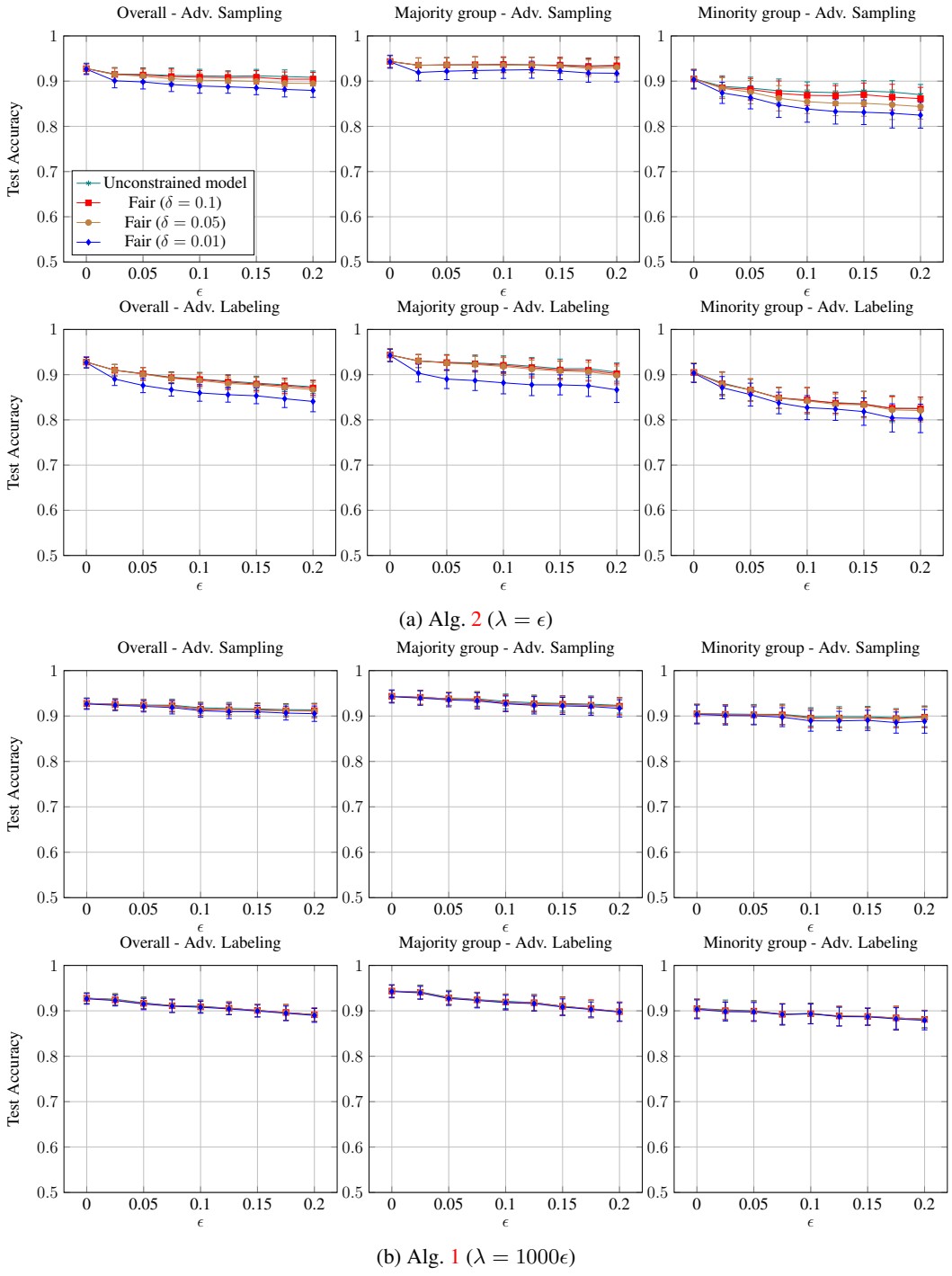

Figure 13: Effect of fairness level $\delta$ on robustness across groups under adversarial sampling and adversarial labeling bias – MEPS dataset. The majority group contributes $58.9\%$ of the test data. The reductions approach (Agarwal et al., 2018) is used to train fair models.

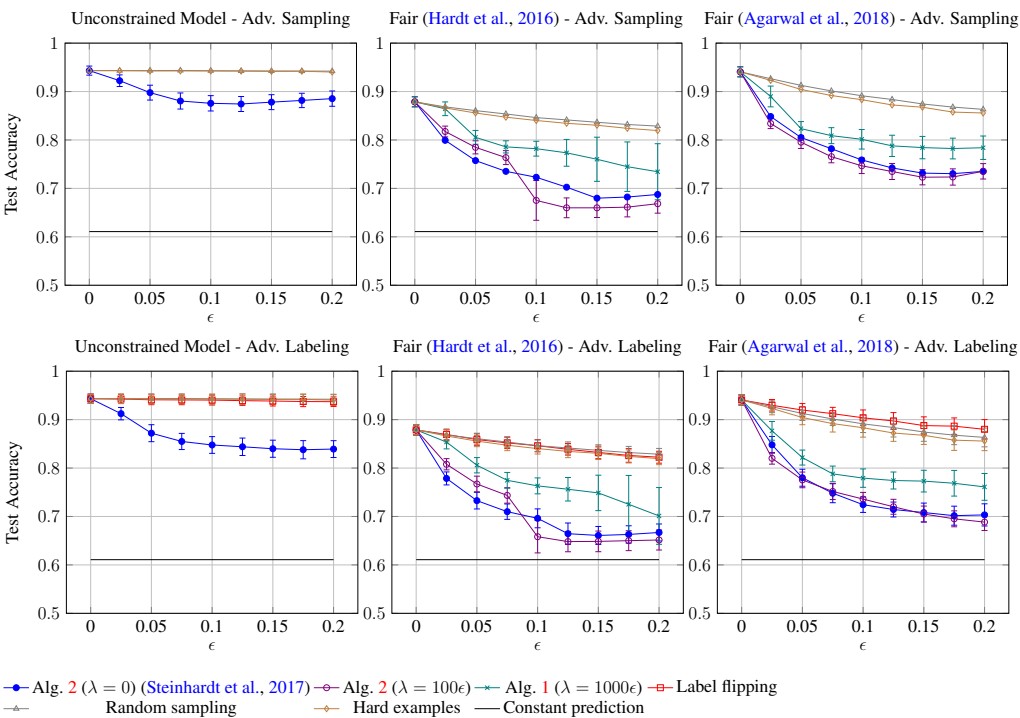

Figure 14: Test accuracy of unconstrained and fair models with respect to equal opportunity in the presence of adversarial bias – COMPAS dataset. The x-axis $\epsilon$ is the ratio of the size of $\mathcal{D}_\mathrm{p}$ to the size of clean dataset $\mathcal{D}_\mathrm{c}$, and reflects the contamination level of the training set. We compare the impact of adversarial bias with baselines and adversarial bias against unconstrained models, for various $\epsilon$. The difference between test accuracy at $\epsilon = 0$ (benign setting) and a larger $\epsilon$ value reflects the impact of the bias. Constant prediction always outputs the majority label in the clean dataset. The enforced fairness level $\delta$ is 0 and 0.01 for the fair model (Hardt et al., 2016) and fair model (Agarwal et al., 2018), respectively.

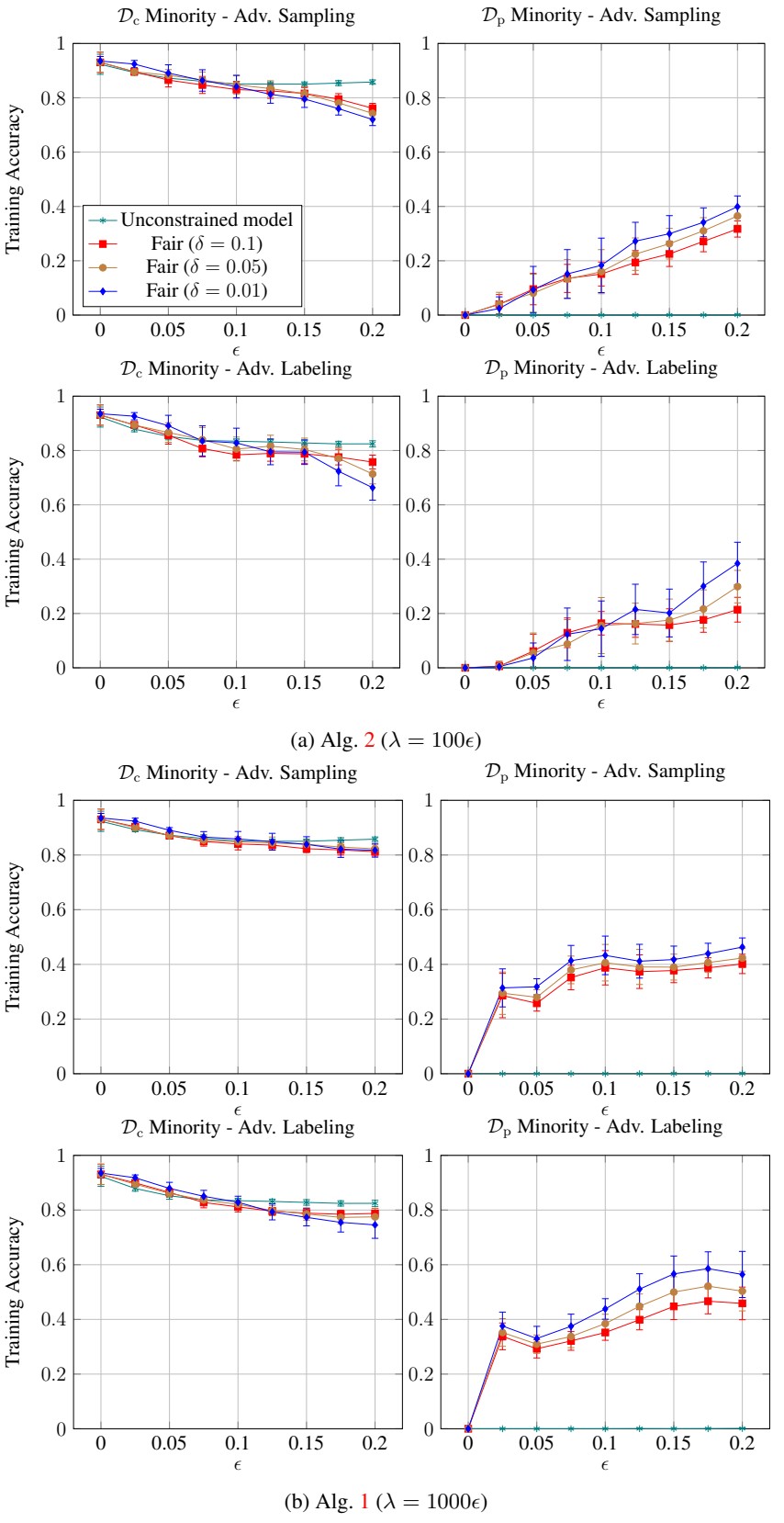

Figure 15: Accuracy of clean training data and $\mathcal{D}_\mathrm{p}$ under adversarial sampling and adversarial labeling bias – COMPAS dataset. The reductions approach Agarwal et al. (2018) is used to train all fair models.

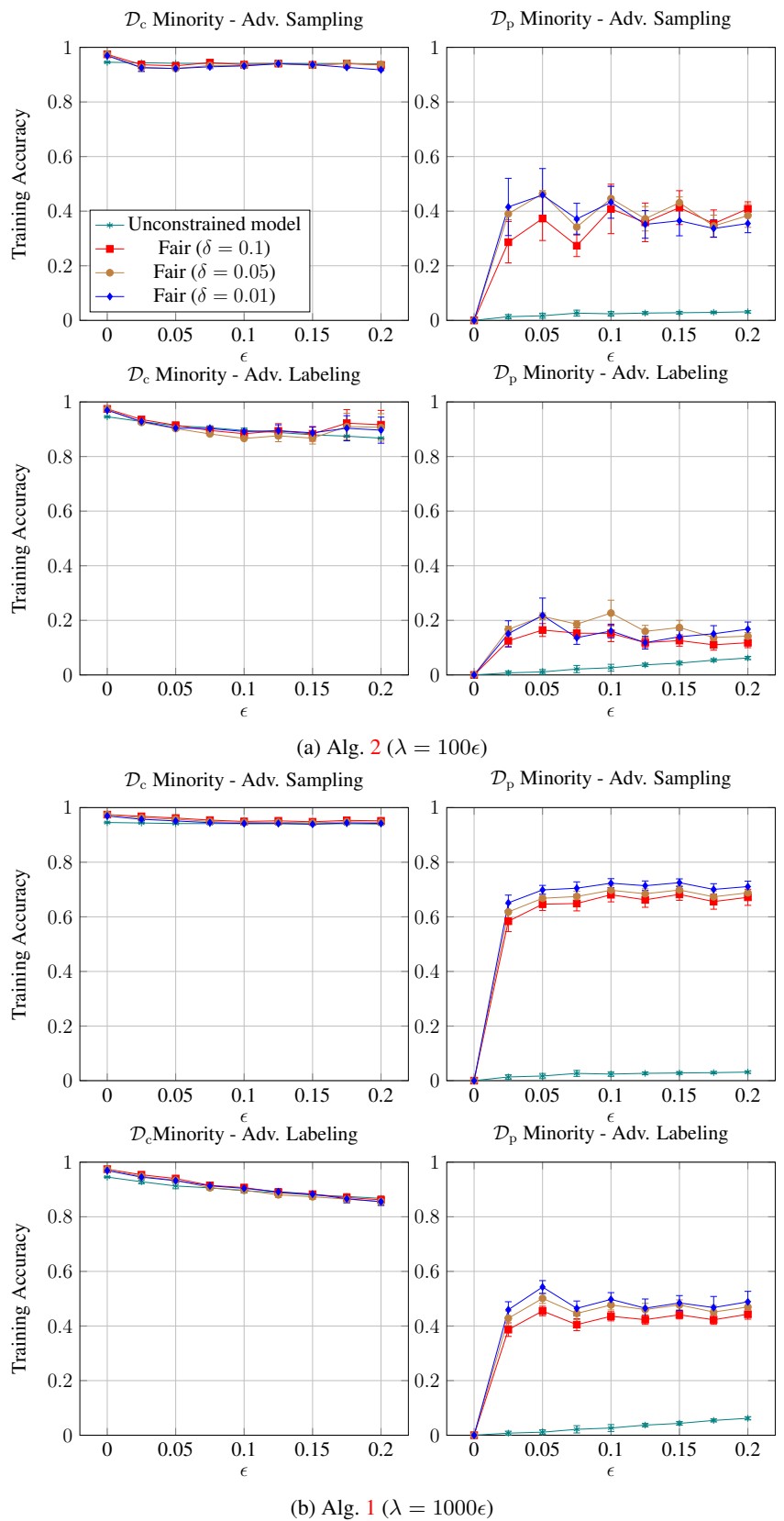

Figure 16: Accuracy of clean training data and $\mathcal{D}_p$ under adversarial sampling and adversarial labeling bias – Adult dataset. Fair models are trained using reduction approach (Agarwal et al., 2018).

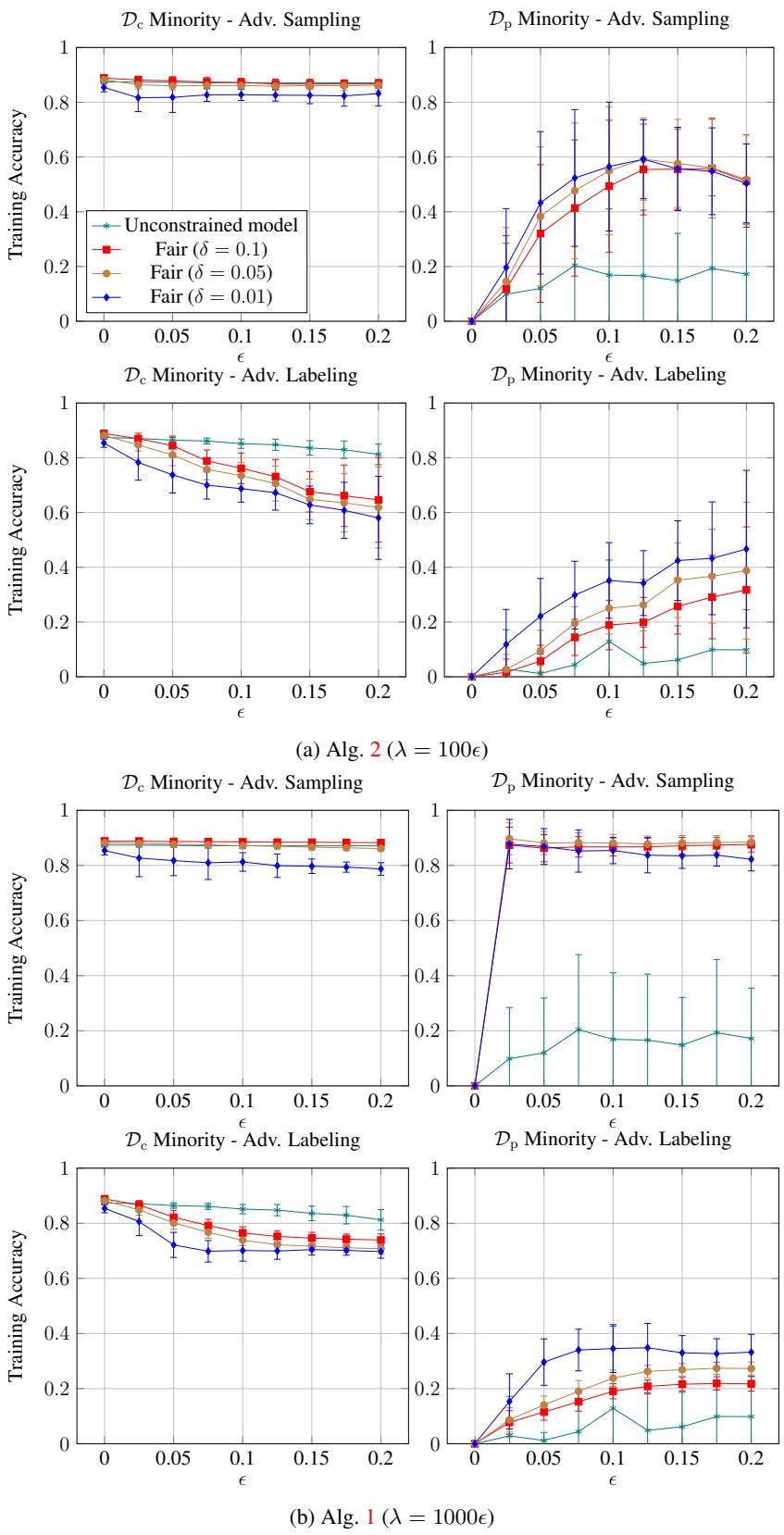

Figure 17: Accuracy of clean training data and $\mathcal{D}_{\mathrm{p}}$ under adversarial sampling and adversarial labeling bias – Synthetic dataset. Fair models are trained using reduction approach (Agarwal et al., 2018).

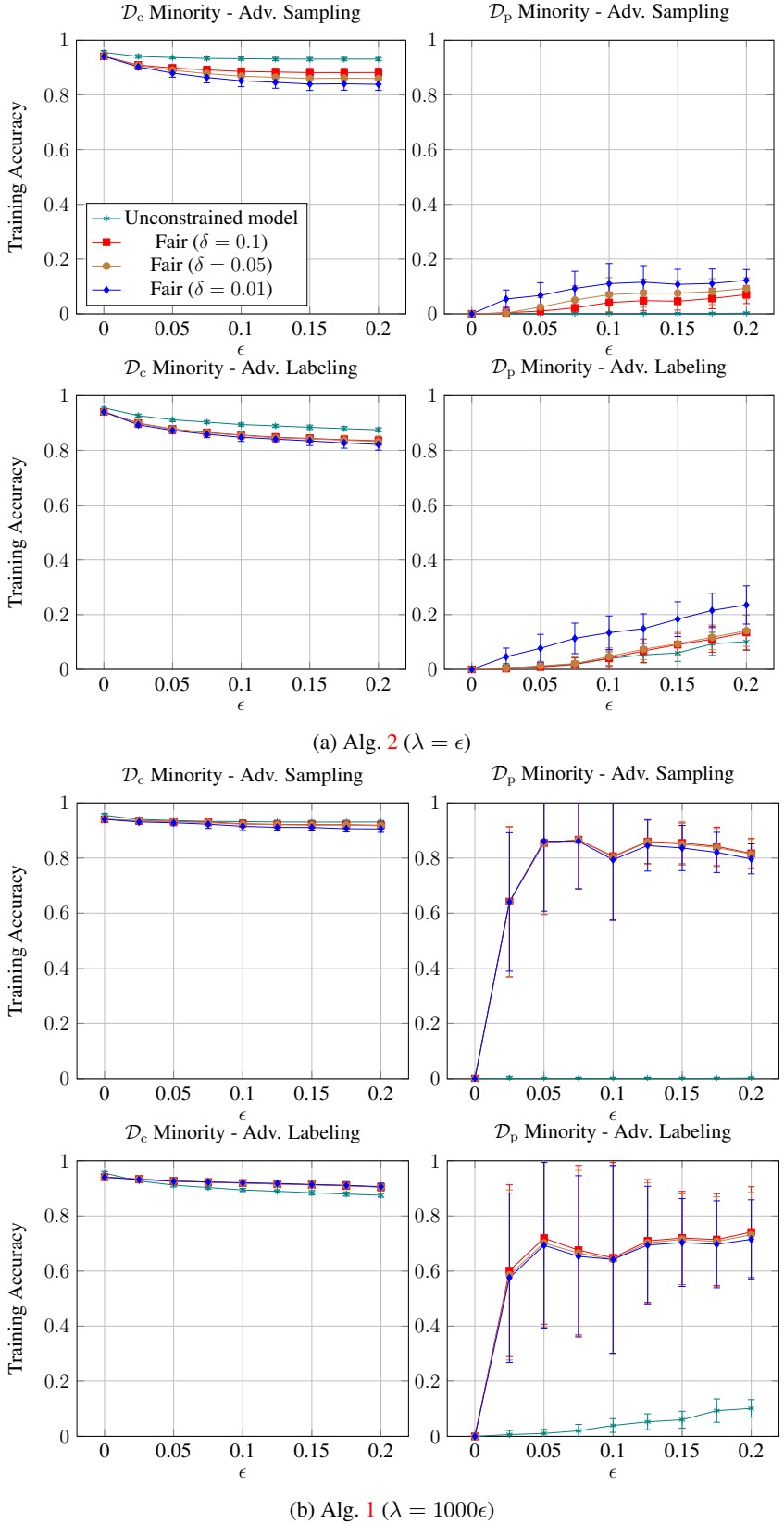

Figure 18: Accuracy of clean training data and $\mathcal{D}_p$ under adversarial sampling and adversarial labeling bias – MEPS dataset. Fair models are trained using reduction approach (Agarwal et al., 2018).

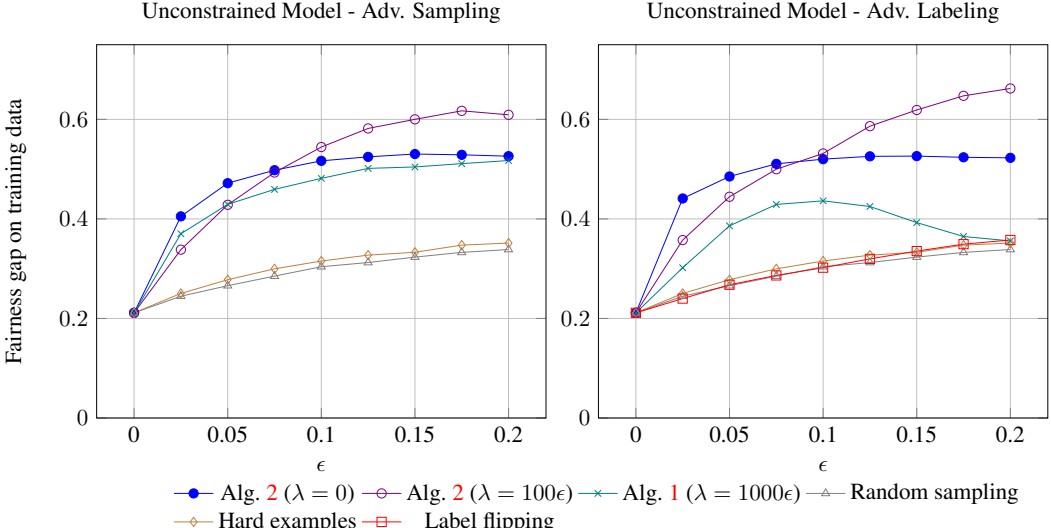

Figure 19: Fairness gap on the unconstrained model with respect to the training data – COMPAS dataset. An unconstrained model is learned on the training data that includes adversarial bias generated by Alg. 2 ($\lambda = 100\epsilon$). The fairness gap $\Delta$ is defined in (1). The numbers reflect how unfair this unconstrained model is with respect to the protected group on the training data.

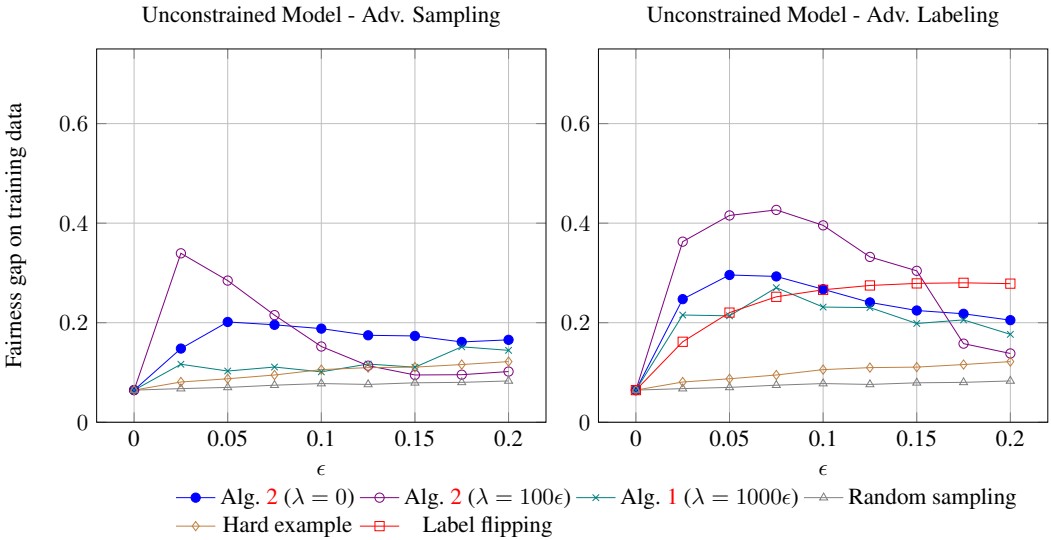

Figure 20: Fairness gap on the unconstrained model with respect to the training data – Adult dataset. An unconstrained model is learned on the training data that includes adversarial bias generated by Alg. 2 ($\lambda = 100\epsilon$). The fairness gap $\Delta$ is defined in (1).

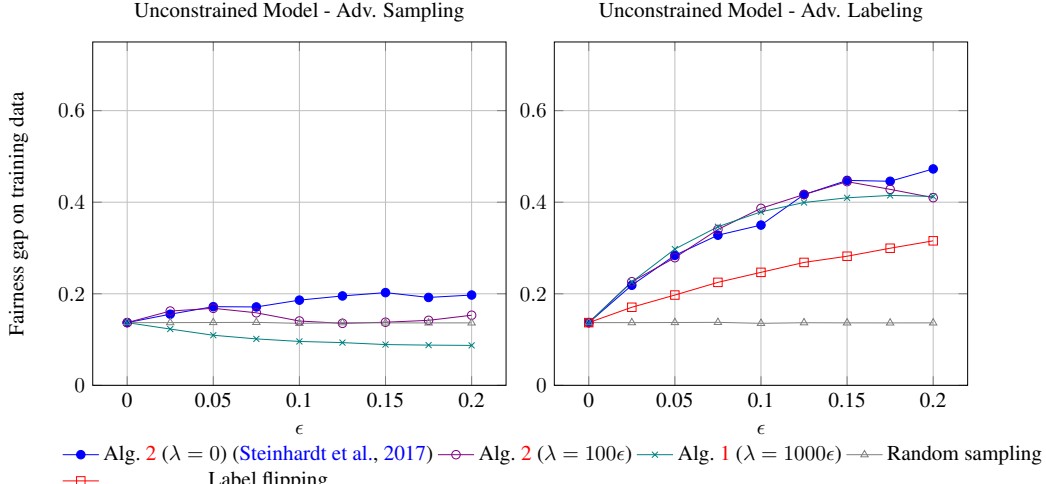

Figure 21: Fairness gap on the unconstrained model with respect to the training data – Synthetic dataset. An unconstrained model is learned on the training data that includes adversarial bias generated by Alg. 2 ($\lambda = 100\epsilon$).

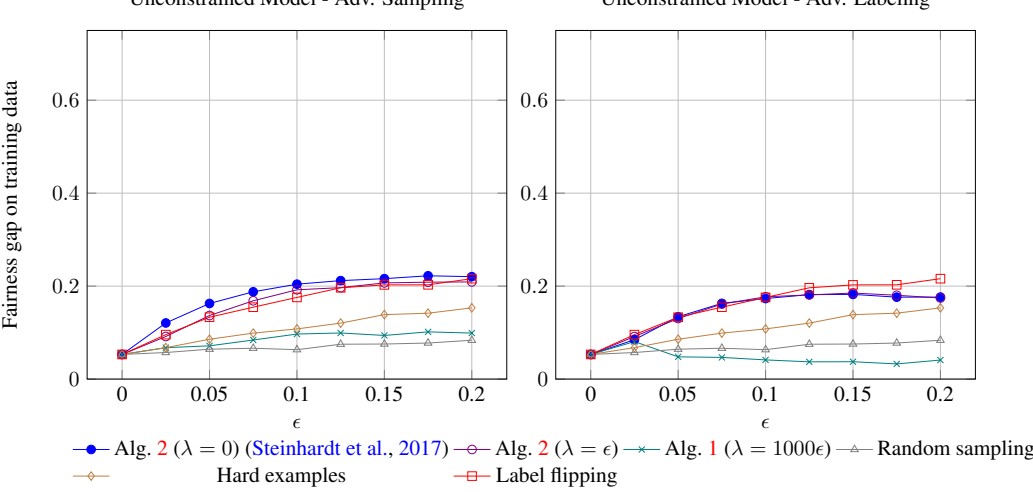

Figure 22: Fairness gap on the unconstrained model with respect to the training data – MEPS dataset. An unconstrained model is learned on the training data that includes adversarial bias generated by Alg. 2 ($\lambda = \epsilon$).

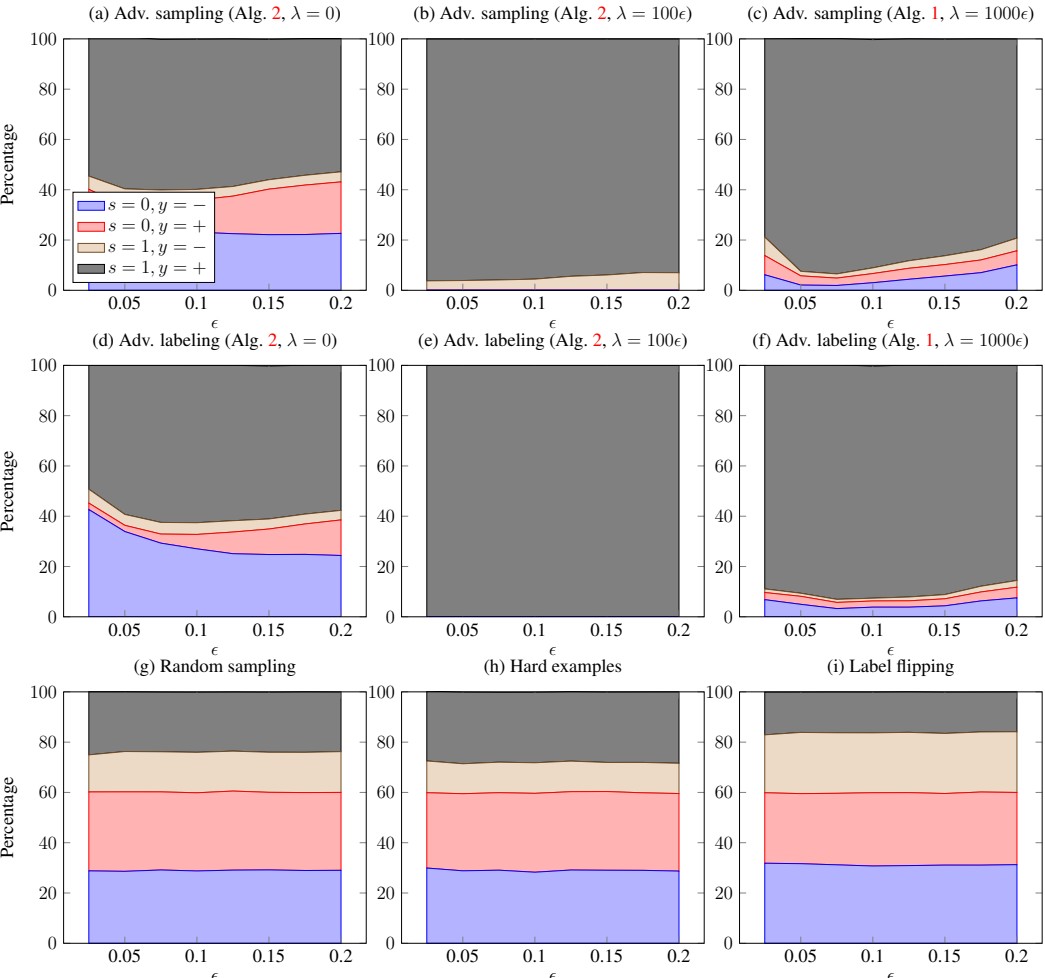

Figure 23: Distribution of $\mathcal{D}_\mathrm{p}$ – COMPAS dataset. We report the protected attribute ($s = 0$ for whites and $s = 1$ for blacks) and labels of data points in $\mathcal{D}_\mathrm{p}$ for various $\epsilon$. For every value of $\epsilon$, the number for each combination of the protected attribute and label reflects the percentage of points with this combination in $\mathcal{D}_\mathrm{p}$. Algorithm 2 with $\lambda = 0$ is the same as the attack algorithm proposed in Steinhardt et al. (2017).

.

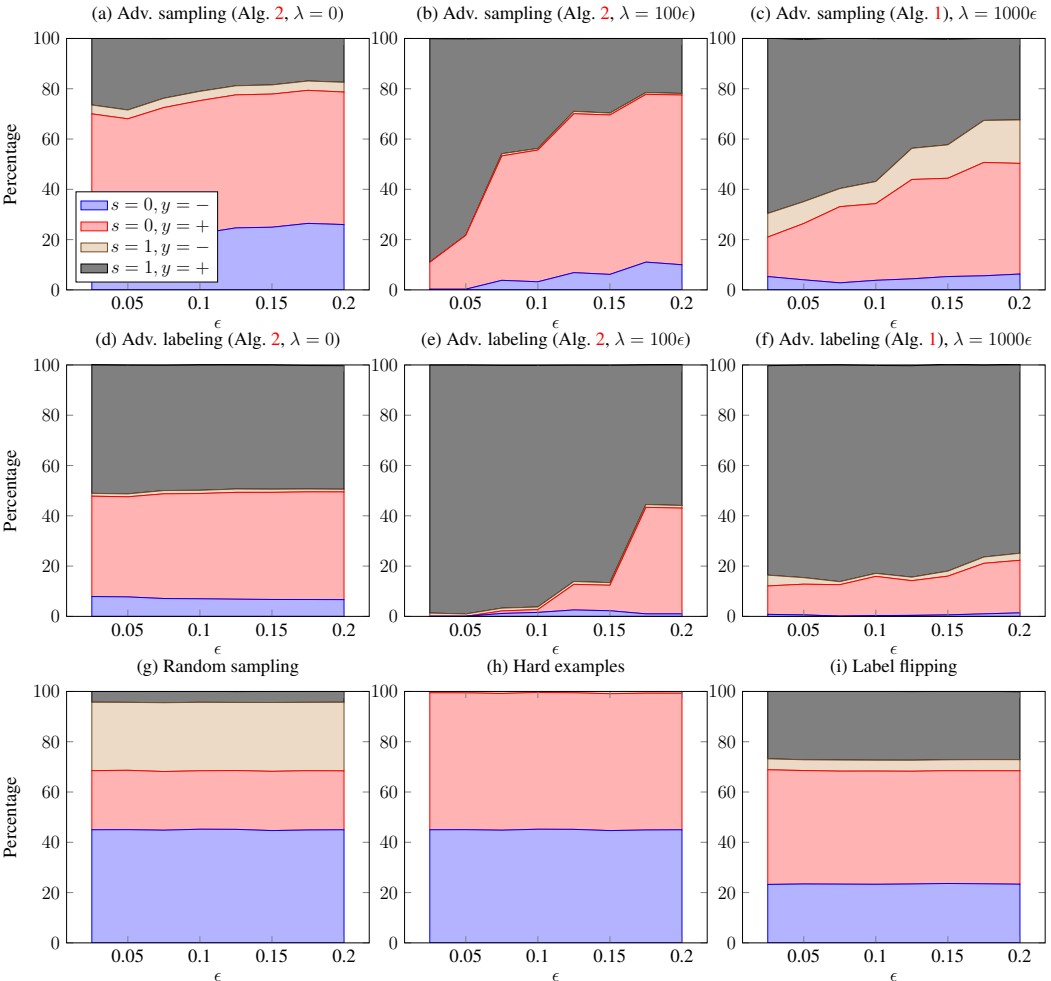

Figure 24: Distribution of $\mathcal{D}_\mathrm{p}$ – Adult dataset. We report the protected attribute ($s = 0$ for males and $s = 1$ for females) and labels of data points in $\mathcal{D}_\mathrm{p}$ for various $\epsilon$. Algorithm 2 with $\lambda = 0$ is the same as the attack algorithm proposed in Steinhardt et al. (2017).

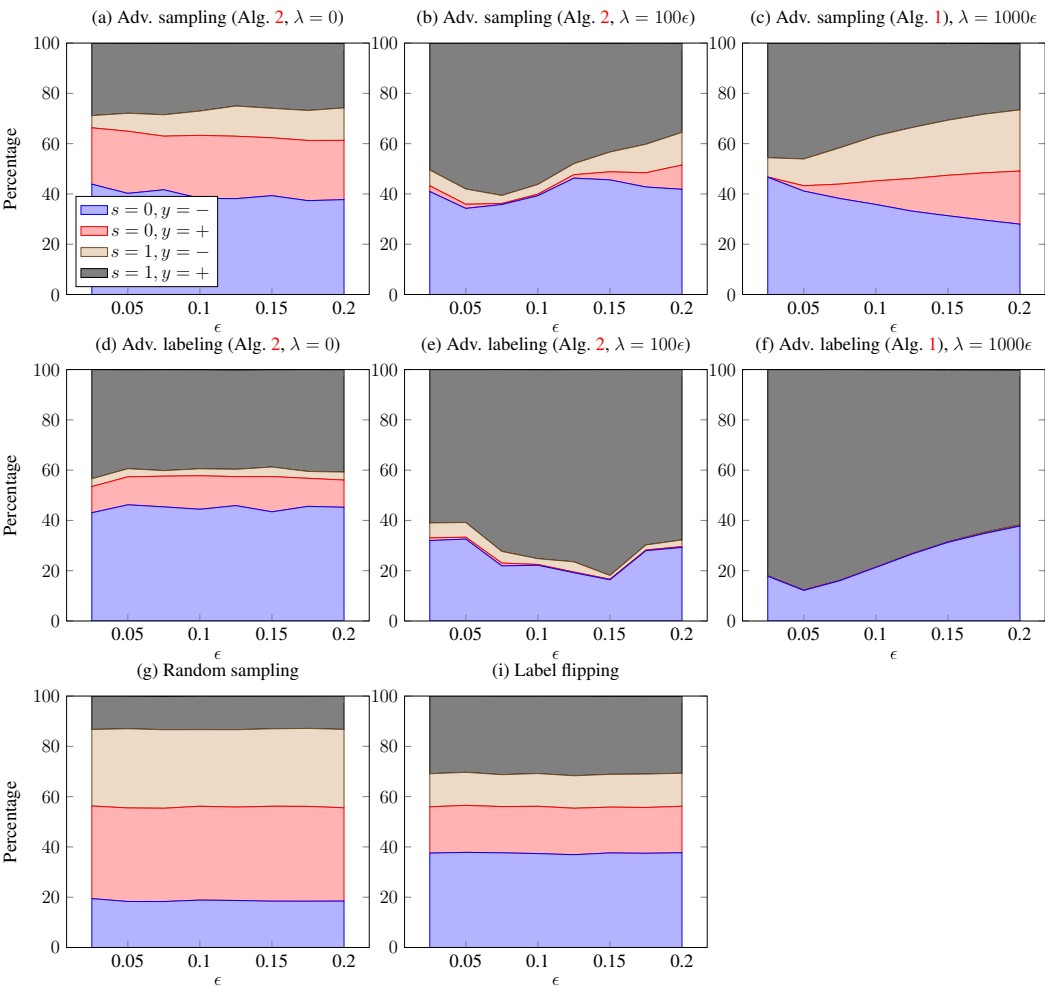

Figure 25: Distribution of the $\mathcal{D}_\mathrm{p}$ – Synthetic dataset. Algorithm 2 with $\lambda = 0$ is the same as the attack algorithm proposed in Steinhardt et al. (2017).

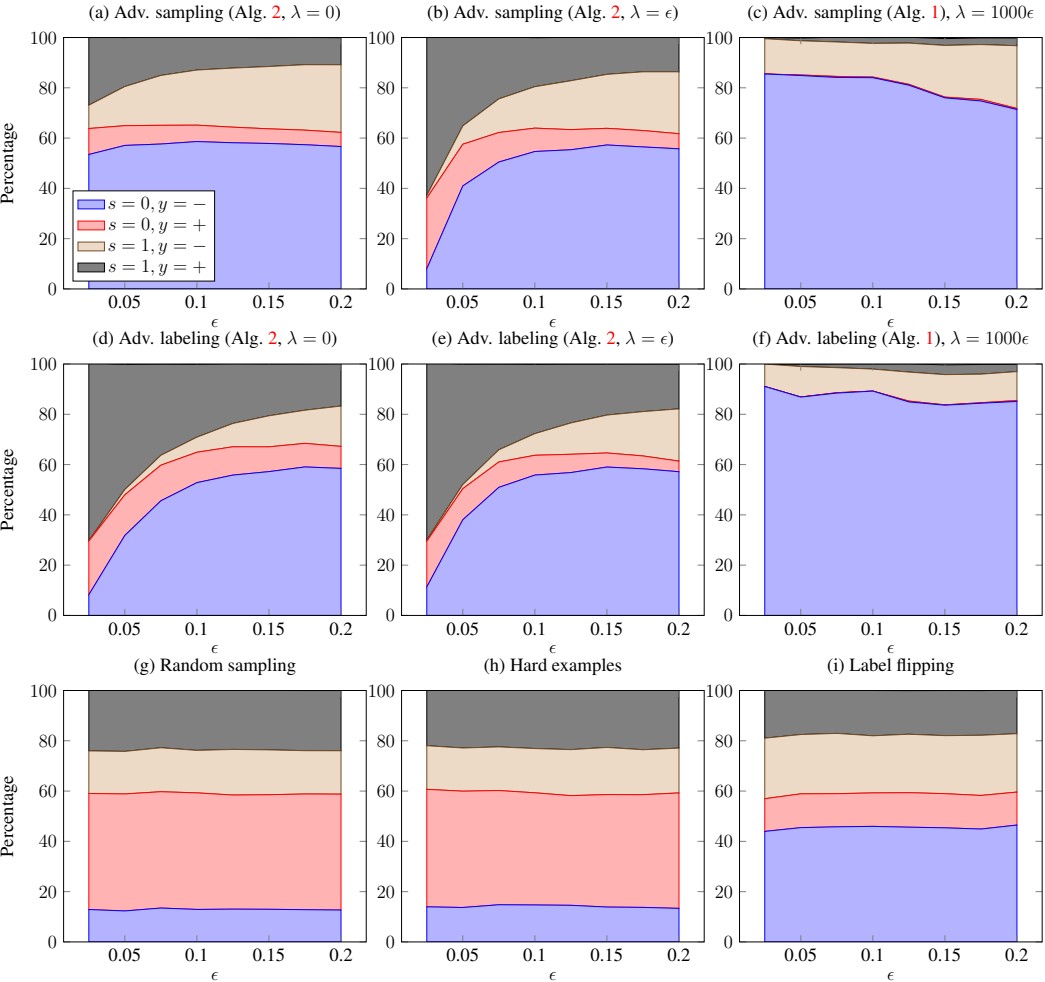

Figure 26: Distribution of the $\mathcal{D}_p$ – MEPS dataset. We report the protected attribute ($s = 0$ for non-white and $s = 1$ for white) and labels of data points in $\mathcal{D}_p$ for various $\epsilon$. For this dataset, $(s = 0, y = -)$ represents the smallest subgroup.

