# OpenReview forum: "On Adversarial Bias and the Robustness of Fair Machine Learning"
_ICLR.cc/2022/Conference — ICLR 2022 Submitted_

### Official Review · Reviewer_oMrk · 2021-10-30

**Correctness:** 3
**Technical Novelty And Significance:** 2
**Empirical Novelty And Significance:** 2
**Recommendation:** 5
**Confidence:** 3

**Main Review:**

Strengths:

1. It investigates an important issue that arises in trustworthy machine learning.
2. Overall the paper is well-written and easy to follow.
3. It provides a theoretical guarantee for the approximated algorithm (Algorithm 1).

Weaknesses:

1. The evaluation is done only under certain algorithms (Algorithms 1 and 2) and under particular fairness algorithms (even assumpting somewhat strict logistic regression models). In addition, the evaluation is based solely upon experiments, although some intuitive remarks are provided along with the experimental results. This reviewer wonders if the key message (fairness degrades accuracy more significantly than the unconstrained model) holds fundamentally. Otherwise, the claim may hold only under certain settings.

2. It considers only one fairness measure, equalized odds. What about another prominent measure like demographic parity?

3. It would be more informative and useful if the challenge that this paper claims can be well addressed by some recent fair-&-robust training algorithms. Actually, if that is the case, I am wondering if the raised issue is really a challenge.


**Summary Of The Paper:**

This paper explores the robustness against certain adversarial train-data poisoning (sampling or label flipping) for some prior fair machine learning algorithms, under equalized odds. Experimental results conducted on some synthetic and benchmark real datasets under logistic regression models demonstrate that the employed fairness constraints yield more performance degradation relative to unconstrained models.

**Summary Of The Review:**

The claims are based mainly on experiments even under some particular algorithms and models. So it is hard to make a scientific judge as to whether the claims are really the case.

---

### Official Review · Reviewer_iCR1 · 2021-11-01

**Correctness:** 4
**Technical Novelty And Significance:** 3
**Empirical Novelty And Significance:** 3
**Recommendation:** 8
**Confidence:** 4

**Main Review:**

Overall, the paper is well written, covers a lot of literature review from recent papers and is a recent and important topic with lots of interesting analysis and experiments.

Strengths:
1. The paper covers an important issue that is also timely.
2. It covers relevant previous work.
3. Lots of experimental results which are interesting.

Weaknesses:
These are not necessarily weaknesses but some points that I think can make the paper more complete and interesting:
1. It would be interesting to see results for other fairness notions like statistical parity and its comparison with equality of op and equalized odds results along with different methods that satisfy statistical parity for instance specifically that are different than methods optimized for equality of op. Would we still see this trend?
2. Is robustness related to fairness notion that we have in mind or the method that we use it to optimize for it? I think these are interesting questions that this type of work can answer by expanding on some experiments and discussion.
3. It would also be interesting to see results from applying existing data poisoning attacks on fairness in addition to the methods authors used in this paper and see how results would fare under those attacks as well...

**Summary Of The Paper:**

In this paper, authors study the trade-offs between fairness and robustness in the adversarial setting. From the results, authors show that fair models are more likely to have lower accuracy in the adversarial setting where fairness is targeted in the training data compared to regular models.

**Summary Of The Review:**

I think the paper has lots of interesting points and its merits outweighs the points I noted under weaknesses which are not really weaknesses but some suggestions that can further improve the work. With that being said, the paper has already enough content for publication which are interesting and informative and the suggestions can maybe considered as a followup work.

---

### Official Review · Reviewer_bW76 · 2021-11-05

**Correctness:** 3
**Technical Novelty And Significance:** 2
**Empirical Novelty And Significance:** 1
**Recommendation:** 3
**Confidence:** 5

**Main Review:**

The strength of the paper is the particular problem considered. I think it is an important problem to make fair models robust before they can be reliably deployed in practice. Additionally, I thought the paper was well-written and I could follow most of the arguments.

Weaknesses:
1. Although the problem of designing fair and robust classifier is interesting, this problem now has been looked into by several papers. I would mention three papers [1,2,3] that were not cited by the paper (in addition to the paper Rezaei et.al. 2020) So, I think the paper does not provide a significant contribution to this problem. In addition, I think some observation made by the authors have already been highlighted by several papers in the past. For example, the fact that adversarial training increases accuracy disparity has been observed by [4].
2. In the experimental section, the choice of fair classifiers is not exhaustive. I would have expected the authors to include at least one pre-processing based classifiers. Furthermore, the choice of datasets is not exhaustive either. If I have to accept the paper based on its empirical observation then it should include additional datasets of different kinds e.g. law-school and communities.
3. In terms of proposed algorithm, I am not sure why the authors chose the current design in algorithm 1. There are many ways to make an existing fair classifier robust. The authors have not provided any justification for the proposed approach.
4. A significant drawback of the algorithm is the linear approximation of the term $\Delta(\theta, \mathcal{D}_c \cup \mathcal{D}_p)$. This assumes that the contribution of each datapoint can be summed independently to get the overall effect. However, I could imagine a setting where new points have very low marginal contribution to the fairness gap when added to a set of $\mathcal{D}_p$ with reasonable number of points.
5. Finally, the related work section of the paper does not provide a detailed description of the latest work. I found several missing references that are quite relevant to the current work.

References:
1. Fair Classification with Adversarial Perturbations (NeurIPS-2021)
2. Ensuring Fairness beyond the Training Data (NeurIPS-2020)
3. Sample Selection for Fair and Robust Training (NeurIPS-2021)
4. Fairness Through Robustness: Investigating Robustness Disparity in Deep Learning (FAccT-2021)

**Summary Of The Paper:**


This paper studies the problem of fair classification when the dataset is adversarially perturbed. In particular, the authors considers two models of adversarial perturbation -- (1) adversarial sampling (outlier data points are chosen adversarially), and (2) adversarial labeling (labels of a fraction of data points are chosen adversarially). Based on no-regret online gradient descent algorithm, the authors propose an algorithm that uses existing fair classifier as an oracle and produces a classifier that is both fair and robust to adversarial perturbations.

The main contribution of the paper is evaluation of robustness of fair classifiers. The main message of the empirical evaluation is the following. First, there is a large relative drop in accuracy for all fair models because of adversarial perturbations. Second, adversarial bias increases the accuracy gap across different groups, and thereby further increases the cost of fairness.

**Summary Of The Review:**

In summary, I think the authors considered an interesting problem. However, the both the proposed algorithm and empirical evaluation have limitations. The main message of the paper is that robustness increases the cost of fairness. This fact has been observed by other papers before. Additionally, in comparison to several recent papers on fair and robust classification, I don't think the paper makes a major contribution to the problem.

---

### Official Review · Reviewer_tNTd · 2021-11-05

**Correctness:** 2
**Technical Novelty And Significance:** 1
**Empirical Novelty And Significance:** 2
**Recommendation:** 3
**Confidence:** 4

**Main Review:**

Major comment:

The paper completely dismisses the recent advances made in the literature.  In particular, the authors ignore (and/or misunderstand) the prior works on "robust fair learning" by writing "Although several fair learning algorithms have been proposed to achieve different robustness properties (Wang et al., 2020; Rezaei et al., 2020; Taskesen et al., 2020; Roh et al., 2020; Cotter et al., 2019), those algorithms do not consider the adversarial bias and fail to be robust against adversarial bias."

Unfortunately, this is clearly false.  To see this, consider [Wang et al. 2020].  Their algorithm and theoretical guarantees do hold for any "perturbation" including adversarial perturbation. More specifically, the only assumption they have on the poisoned data distribution is that it is within a certain Total Variation distance from the true distribution, which does not distinguish random noise and adversarial noise.

Another instance is [Roh et al. 2020], which indeed performed an adversarial label flipping attack proposed in ["Label sanitization against label flipping poisoning attacks", Paudice et al.]. They showed that the fair learning algorithms can be made robust against such adversarial data poisoning attacks. Also, the same observation that "fair learning algorithms are more vulnerable than vanilla training algorithms" was also made in this work.

Given these prior studies on robust fair learning algorithms, the authors should have discussed how their proposed attack algorithms perform against such robust fair learning algorithms.


Other comments:

-- The algorithmic novelty is marginal: the proposed algorithm is an immediate application of Steinhardt et al. with a different objective function.

-- It is also unclear how effective the proposed approximation for the fairness penalty (shown in (4)) is.  The authors may want to report how good this approximation is.  In particular, for a fixed dataset, one can measure the original fairness penalty value and the proposed approximation.  By repeating these measurements for various classifiers, one can see how these two values are related by plotting a scatter plot.

-- All the tables and figures show how much test accuracy drops.  What about their fairness violations?  Also, what happens when one simply performs an attack based on the prediction loss term alone without considering the fairness constraint term?  How does the attack performance change?

**Summary Of The Paper:**

Intuitively, the same amount of data poisoning will have a larger impact when the learner is solving a constrained optimization problem than an unconstrained one.  This paper proposed an attack algorithm specifically designed for such fair learners and showed that fair learning algorithms are more vulnerable to adversarial data poisoning attacks.  The problem studied in this paper is of high importance.

**Summary Of The Review:**

The paper studies an interesting problem.  However, it is unclear whether their proposed attack algorithm is still valid given that researchers have already proposed various robust fair learning algorithms.

---

### Decision · Program_Chairs · 2022-01-20

**Decision:**

Reject

**Comment:**

The authors consider the impact of designing fair algorithms on adversarial robustness. The particular focus is on poisoning attacks. They show experimentally that for some datasets and models/algorithms that using "fair" algorithms increase adversarial vulnerability compared to the standard training procedure (ignoring fairness criteria). The reviews have raised questions about whether the experimental results are extensive enough and I share their concerns. Most importantly, the authors have not addressed the question regarding to the quality of approximation at all.